# ACK1 and BRK non-receptor tyrosine kinase deficiencies are associated with familial systemic lupus and involved in efferocytosis

Stephanie Guillet[1,2†], Tomi Lazarov[1,3*†], Natasha Jordan[4§], Bertrand Boisson[5,6], Maria Tello[1], Barbara Craddock[7], Ting Zhou[7], Chihiro Nishi[8], Rohan Bareja[9], Hairu Yang[1], Frederic Rieux-Laucat[6], Rosa Irene Fregel Lorenzo[10], Sabrina D Dyall[11], David Isenberg[12], David D'Cruz[4], Nico Lachmann[13], Olivier Elemento[9], Agnes Viale[14], Nicholas D Socci[14,15], Laurent Abel[5,6], Shigekazu Nagata[8], Morgan Huse[1], W Todd Miller[15,16‡], Jean-Laurent Casanova[5,6,17,18,19‡], Frédéric Geissmann[1,3,4*]

*For correspondence:
lazarovt@mskcc.org (TL);
geissmaf@mskcc.org (FG)

[†]These authors contributed equally to this work
[‡]These authors also contributed equally to this work

Present address: [§]Rheumatology Department, Cambridge University Hospitals, Cambridge, United Kingdom

Competing interest: The authors declare that no competing interests exist.

[1]Immunology Program, Sloan Kettering Institute, Memorial Sloan Kettering Cancer Center, New York, United States; [2]Ecole doctorale Bio Sorbonne Paris Cité, Université Paris Descartes-Sorbonne Paris Cité, Paris, France; [3]Immunology and Microbial Pathogenesis Program, Weill Cornell Graduate School of Medical Sciences, New York, United States; [4]Centre for Molecular and Cellular Biology of Inflammation (CMCBI), King's College London and Louise Coote Lupus Unit, Guy's and Thomas' Hospitals, London, United Kingdom; [5]St. Giles Laboratory of Human Genetics of Infectious Diseases, Rockefeller Branch, The Rockefeller University, New York, United States; [6]University of Paris Cité, Imagine Institute, Paris, France; [7]SKI Stem Cell Research Core, Memorial Sloan Kettering Cancer Center, New York, United States; [8]Laboratory of Biochemistry & Immunology, World Premier International Immunology Frontier Research Center, Osaka University, Osaka, Japan; [9]Cary and Israel Englander Institute for Precision Medicine, Institute for Computational Biomedicine, Meyer Cancer Center Weill Cornell Medical College, New York, United States; [10]University of La Laguna, San Cristóbal de La Laguna, Spain; [11]Department of Biosciences and Ocean Studies, Faculty of Science, University of Mauritius, Reduit, Mauritius; [12]Bioinformatics Core, Memorial Sloan Kettering Cancer Center, New York, United States; [13]Centre for Rheumatology, Division of Medicine, University College London, The Rayne Building, London, United Kingdom; [14]Institute of Experimental Hematology, REBIRTH Cluster of Excellence, Hannover Medical School, Hannover, Germany; [15]Marie-Josée & Henry R. Kravis Center for Molecular Oncology, Memorial Sloan Kettering Cancer Center, New York, United States; [16]Department of Physiology and Biophysics, Stony Brook University School of Medicine, Stony Brook, United States; [17]Howard Hughes Medical Institute, New York, United States; [18]Lab of Human Genetics of Infectious Diseases, INSERM, Necker Hospital for Sick Children, Paris, France; [19]Department of Pediatrics, Necker Hospital for Sick Children, Paris, France

### eLife assessment

In this **important** study, the authors found, with the use of statistical methods, that compound heterozygous rare deletion variants affecting the kinase-domain of non-receptor tyrosine kinase TNK2/ACK1 and PTK6/BRK are associated with human systemic lupus erythematosus (SLE). The authors use a **convincing** mouse experimental model and human-induced pluripotent stem cell (hiPSC)-derived macrophages to clarify cause-effect relationships and the cellular basis of nephritis. With the identification of new SLE-related genes, this manuscript improves our understanding of human SLE pathogenesis.

**Abstract** Systemic lupus erythematosus (SLE) is an autoimmune disease, the pathophysiology and genetic basis of which are incompletely understood. Using a forward genetic screen in multiplex families with SLE, we identified an association between SLE and compound heterozygous deleterious variants in the non-receptor tyrosine kinases (NRTKs) *ACK1* and *BRK*. Experimental blockade of ACK1 or BRK increased circulating autoantibodies in vivo in mice and exacerbated glomerular IgG deposits in an SLE mouse model. Mechanistically, NRTKs regulate activation, migration, and proliferation of immune cells. We found that the patients' *ACK1* and *BRK* variants impair efferocytosis, the MERTK-mediated anti-inflammatory response to apoptotic cells, in human induced pluripotent stem cell (hiPSC)-derived macrophages, which may contribute to SLE pathogenesis. Overall, our data suggest that ACK1 and BRK deficiencies are associated with human SLE and impair efferocytosis in macrophages.

### Introduction

Systemic lupus erythematosus (SLE) is a chronic autoimmune rheumatic disease, characterized by the presence of circulating autoantibodies against nuclear antigens. Clinical manifestations vary among affected individuals and can involve many organs and systems, including the skin, joints, kidneys, heart, lungs, nervous system, and hematopoietic system (*Dall'Era, 2013*; *Petri et al., 2012*). The prevalence of SLE ranges from 0.4 to 2/1000, varying with sex, age, and ancestry, and is more common in women of childbearing age and individuals of African, Asian, and Hispanic ancestry (*Dall'Era, 2013*; *Petri et al., 2012*; *Rees et al., 2017*; *Johnson et al., 1995*; *Feldman et al., 2013*; *Somers et al., 2014*; *Lim et al., 2014*; *Ferucci et al., 2014*; *Chakravarty et al., 2007*). Sex, hormones, and environmental factors, including drugs and chemical exposures, viral infections, and sunlight, contribute to the disease. Currently, specific therapies for SLE are limited (*Hoi and Morand, 2021*), and clinical manifestations such as lupus nephritis, one of the most common and serious manifestations of SLE, remain a major risk factor for morbidity and mortality (*Tektonidou et al., 2016*; *Menez et al., 2018*; *Hanly et al., 2016*; *Parikh et al., 2020*). The contribution of genetics to SLE is supported by epidemiological data showing familial aggregation (*Arnett and Shulman, 1976*; *Alarcón-Segovia et al., 2005*; *Reichlin et al., 1992*) and higher concordance rates between monozygotic than dizygotic twins (*Deapen et al., 1992*), the association of autosomal recessive deficiency in *PKCdelta* or *DNAse1L3* with familial SLE, and similar phenotypes in the corresponding mouse models (*Belot et al., 2013*; *Kiykim et al., 2015*; *Salzer et al., 2013*; *Al Mayouf et al., 2011*; *Ozçakar et al., 2013*; *Sisirak et al., 2016*). In addition, genome wide association studies in large populations of patients have implicated a number of genes associated with immune system function (*Dall'Era, 2013*; *Teruel and Alarcón-Riquelme, 2016*), and an SLE-like disease is observed in a proportion of patients with immunodeficiency due to autosomal recessive or X-linked *C1q*, C1r/s, *C2*, *C4*, and *NADPH-Oxidase* deficiencies (*Pickering et al., 2000*; *Winkelstein et al., 2000*; *Foster et al., 1998*; *van den Berg et al., 2009*; *Ling et al., 2018*), with Rasopathies due to autosomal dominant gain-of-function mutations in the RAS pathway (*Bader-Meunier et al., 2013*), and a proportion of patients presenting with interferonopathies with bi-allelic or mono-allelic mutations in genes coding for nucleic acid sensors such as *TREX1*, *STING*, *SAMHD1*, *ADAR*, and *IFIH1* (*Jeremiah et al., 2014*; *Crow et al., 2015*; *Lee-Kirsch et al., 2007*; *Ramantani et al., 2011*).

In this study, we performed a forward genetic screen in multiplex SLE families with a well-defined phenotype, lupus nephritis, to identify new SLE causing genes and molecular pathways involved in

SLE. We report the characterization of novel and rare deleterious variant alleles of two genes encoding the NRTK tyrosine kinase non-receptor 2/activated CDC42 kinase 1 (*TNK2/ACK1*) and protein tyrosine kinase 6/breast tumor kinase (*PTK6/BRK*) in two multiplex families. The variant alleles strongly decrease ACK1 and BRK kinase activity. ACK1 and BRK were shown to control B-cell and T-cell proliferation, survival, and activation (*Kasprzycka et al., 2006*; *Sridaran et al., 2022*). Although *ACK1*- and *BRK*- genetic deficiency (*Sridaran et al., 2022*; *Haegebarth et al., 2006*) or their pharmacological inhibition does not result in spontaneous lupus development in C57BL/6 mice, we show that ACK1 or BRK inhibitors aggravate IgG glomerular deposition in the kidneys of BALB/cByJ mice treated with pristane to induce a lupus-like disease (*Satoh and Reeves, 1994*) and increases serum autoantibody levels. NRTKs mediate phosphorylation of downstream effectors, including RAC1, AKT, and STAT1/3, which are involved in immune cells homeostasis, and their deficiency can cause autoimmunity through different mechanisms (*Yu et al., 2001*.) NRTKs such as ACK1 (*Mahajan et al., 2005*), Src (*Yi et al., 2009*), and PTK2/FAK (*Wu et al., 2005*; *Tibrewal et al., 2008*) are also targets of MERTK, which mediates efferocytosis, the recognition of phosphatidylserine (PtdSer) on apoptotic cells for their anti-inflammatory engulfment (*Wanke et al., 2021*; *Scott et al., 2001*; *Seitz et al., 2007*; *Cohen et al., 2002*). MERTK deficiency is a cause of SLE-like disease (*Cohen et al., 2002*). We found that the patients' ACK1 and BRK variants are kinase dead in response to MERTK activation and fail to phosphorylate AKT and STAT3, and to activate RAC1. Human induced pluripotent stem cell (iPSC)-derived macrophages from patients and isogenic variants presented with defective AKT/STAT3-driven anti-inflammatory response and control of TNF and IL1β production in response to apoptotic cells and had a modest decrease in uptake of apoptotic cells, in comparison to familial and isogenic controls. In contrast, ACK1 and BRK kinase activity are dispensable for the phagocytosis of polystyrene beads, opsonized cells, and microbes. These results altogether suggest that ACK1 and BRK deficiencies underlie SLE in the two families, and that a defective efferocytic response to apoptotic cells may contribute to the auto-immune phenotype of the patients.

## Results

### NRTK compound heterozygous missense variants in two multiplex families with SLE

We recruited 10 multiplex SLE families, each with two or three individuals diagnosed with biopsy confirmed lupus nephritis, classified according to the SLICC criteria (*Petri et al., 2012*) for whom we obtained genomic DNA (blood) in the Louise Coote Lupus Clinic at Guy's and St Thomas' Hospitals, London. We collected peripheral blood from a total of 22 patients and 17 relatives. Thirty five percent (%) of patients were male, and the same proportion were diagnosed before the age of 18 years. Genomic DNA was submitted to whole exome sequencing. Polymorphisms with a minor allele frequency (MAF) >0.01 in the publicly available database gnomAD (120,000 individuals), 1000 genome Project (2504 individuals) and our in-house database (>10,000 individuals) were excluded from analysis. We analyzed each kindred independently, under X-linked recessive, autosomal dominant, or autosomal recessive models of inheritance, and this analysis identified candidate genes in two kindreds.

In Family 1, we identified compound heterozygous missense variants in the non-receptor tyrosine kinase (NRTK) *ACK1*, which were confirmed by Sanger sequencing (*Figure 1A*, *Figure 1—figure supplement 1*). The two patients were males and developed a class IV lupus nephritis between age 10 and 15. The K161Q allele was inherited from one parent and the A156T allele from the other (*Figure 1A*). Principal components analysis (PCA) based on the whole-exome sequencing to analyze population structure, parental inbreeding, and familial linkage (*Belkadi et al., 2016*) reveals South Asian ancestry, the closest 1,000 Genomes Project individuals being those from North India (India, Bangladesh, and Pakistan, see Materials and methods). The *ACK1* mutant alleles, A156T and K161Q (transcript ENST00000333602, *Figure 1—figure supplement 1*), have not been reported in South Asian (31,442 alleles) or other populations from public database (gnomAD, 1000 genomes project) or our own in-house databases of >10,000 exomes of patients with infectious phenotypes. Finally, these variants were not found in DNA from 100 individuals from the small southeastern island from which they originated (see Materials and methods). Altogether, *ACK1* mutant alleles, A156T and K161Q

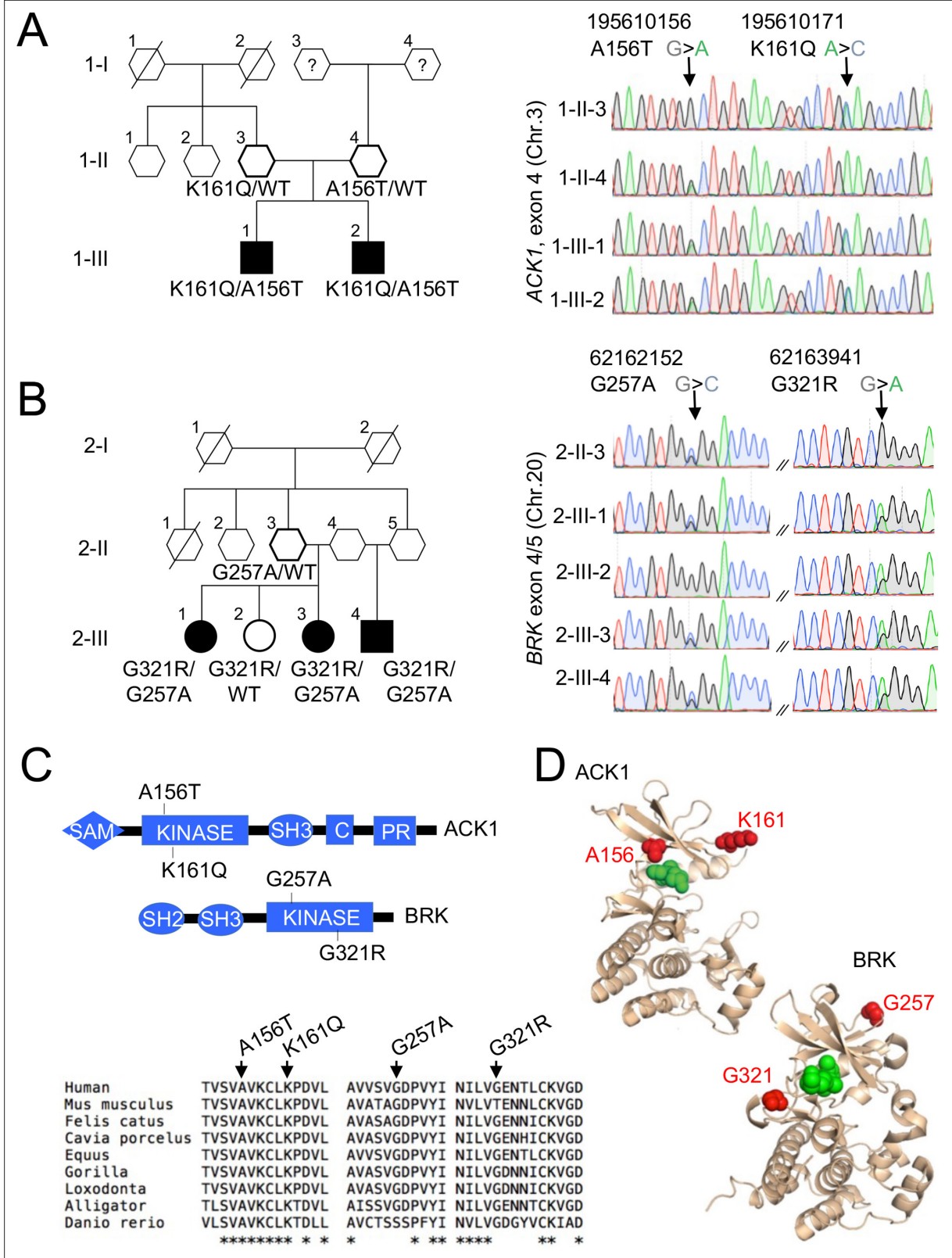

**Figure 1.** NRTK compound heterozygous missense variants in two multiplex families with SLE. (**A, B**) Pedigrees and Sanger re-sequencing of DNA from patients and healthy relatives of kindred 1 (**A**) carrying K161Q and A156T *ACK1* mutations and kindred 2 (**B**) carrying G257A and G321R *BRK* mutations. Individuals with SLE are indicated by black shapes; deceased individuals are shown by a diagonal bar; shapes with thick outline indicate the members analyzed by WES; squares indicate males, circles indicate females, and hexagons indicate generation I or II individuals with undisclosed

*Figure 1 continued on next page*

*Figure 1 continued*

sex for confidentiality. Black: Guanine, green: Adenine, red: Thymidine, blue: Cytosine. Arrows indicate nucleotide substitutions. Text indicates amino-acid substitutions. (**C**) Domain architecture(top panel) of *ACK1* and *BRK*, with indicated mutations. SH2, Src homology 2; SH3, Src homology 3; Kinase, tyrosine kinase domain; C, Cdc42 binding domain; PR, Proline rich domain; SAM, Sterile α motif. Alignment of kinase domains (bottom panel) from ACK1 and BRK orthologs. Arrows indicate positions of mutations and stars indicate the amino acids conserved throughout species. (**D**) Three dimensional (3D) structures of ACK1 and BRK. *Top*: the crystal structure of ACK1 in a complex with AMP-PCP (PDB ID: 1U54). The mutated residues (A156 and K161) are shown in red, and the nucleotide analog is in green. *Bottom*: the crystal structure of BRK in a complex with the ATP-competitive inhibitor dasatinib (PDB ID: 5H2U). The mutated residues (G257 and G321) are shown in red, and dasatinib is in green.

The online version of this article includes the following figure supplement(s) for figure 1:

**Figure supplement 1.** Candidate genes identified by WES analysis in family 1 and family 2.

**Figure supplement 2.** Haplotype member's family 2.

**Figure supplement 3.** SLE-causing genes.

are private to this family and their segregation is compatible with an autosomal recessive trait with complete penetrance.

In Family 2, we identified compound heterozygous missense variants in another NRTK, *BRK* in three siblings (*Figure 1B*, *Figure 1—figure supplement 1*). Two patients developed lupus nephritis, and the third patient developed a severe panniculitis, between the ages of 20 and 30 years. Two patients were female and one male. One parent, who was not genotyped, probably transmitted the G321R to their children, and the 2 other parents (who are also siblings) transmitted the G257A allele (only one parent was available for genotyping; *Figure 1B*, *Figure 1—figure supplement 2*). An unaffected sibling (2.III.2) carries the G321R allele but not the G257A allele (*Figure 1B*, *Figure 1—figure supplement 2*). PCA analysis showed that individuals from family 2 have Sub-Saharan African ancestry, with the closest populations of 1000 genomes project being African Caribbean in Barbados and African Ancestry in Southwest US and Luhya in Webuye in Kenya (see Materials and methods). These 2 *BRK* mutant alleles, G257A and G321R are reported with a maximum MAF of ~$8 \times 10^{-5}$ and $5 \times 10^{-3}$ respectively in Sub-Saharan African subpopulations (*Figure 1—figure supplement 1*), predicting a homozygosity frequency of ~$10^{-8}$ and $10^{-6}$, respectively. These alleles are extremely rare outside Africa in the non-African gnomAD populations. These results are fully consistent under a recessive model with the overall prevalence of SLE (40–200 per 100,000 individuals). Genes that may cause SLE with high or low penetrance in accordance with their inheritance mode (*Figure 1—figure supplement 3*) were not candidates in these kindreds.

## The patients' ACK1 and BRK variants are kinase null and hypomorphic

The ACK1 K161Q A156T and BRK G257A and G321R variants are all localized in the evolutionarily conserved kinase domains of the two proteins (*Figure 1C*), within the N-terminal (ATP-binding) lobes of the kinase catalytic domains (*Lougheed et al., 2004*; *Thakur et al., 2017*; *Figure 1D*), and near the positions of other mutations that decrease kinase activity (*Prieto-Echagüe et al., 2010b*; *Tsui and Miller, 2015*; *Figure 1C*). The variants are all predicted to be deleterious based on combined annotation dependent depletion (CADD) corrected with mutation significance cutoffs (MSC; *Kircher et al., 2014*; *Itan et al., 2016*; *Figure 1—figure supplement 1*). To examine the functional effects of the mutations, we first expressed mutant forms of ACK1 and BRK in HEK293T cells. In vitro kinase assays indicated that ACK1 A156T and BRK G321R are kinase-dead mutants, and that ACK1 K161Q and BRK G257A are severe hypomorphs with an ~*20%* residual kinase activity in vitro (*Figure 2A*). ACK1 A156T and K161Q and BRK G321R variants also lacked auto-phosphorylation activity (*Figure 2B*), phenocopying the effect of the specific ACK1 and BRK kinase inhibitors AIM100 (*Mahajan et al., 2012*) and Cpd4f (*Oelze et al., 2015*) respectively (*Figure 2C*), while the BRK G257A allele has a small residual activity (*Figure 2C*). To examine the effects of the mutations in the patient's cells, we generated iPSCs from unrelated WT donors, patient 1-III-1 (ACK1 mutant), his heterozygous parent 1-II-3, patient 2-III-3 (BRK mutant) and her heterozygous sibling 2-III-2 (*Figure 2—figure supplement 1A–D*), and differentiated them into iPSC-derived macrophages (*Lachmann et al., 2015*: *Figure 2D*; *Figure 2—figure supplement 1E*). hiPSC-derived macrophages from unrelated donors, familial controls, and patients presented with a normal morphology, survival, and phenotype and expressed comparable amount of ACK1 and BRK at the transcript and protein level (*Figure 2D and E*, *Figure 2—figure supplement 1G*), but an in vitro kinase assay indicated a loss of ACK1 kinase activity in ACK1$^{K161Q/A156T}$ macrophages

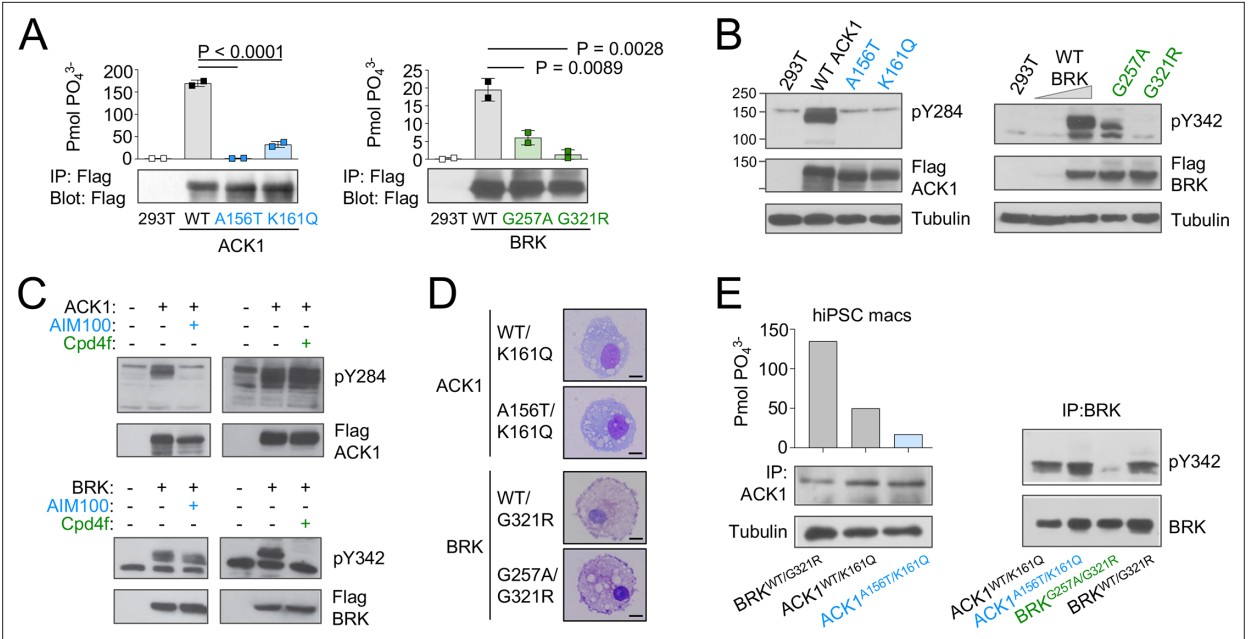

**Figure 2.** ACK1 and BRK mutations are null and hypomorph alleles. (**A**) Immunoprecipitation (IP) kinase assay. ACK1 (left panel) was immunoprecipitated from 293T cells expressing Flag-tagged ACK1 wild type (WT), ACK1 A156T, or ACK1 K161Q with anti-Flag Ab. Immunoprecipitated proteins were used for duplicate in vitro kinase reactions with WASP synthetic peptide. Samples of the immunoprecipitates were also probed with anti-Flag Ab. BRK (right panel) was immunoprecipitated as above from 293T cells expressing Flag-tagged BRK WT and mutants with anti-Flag Ab. Kinase reactions were performed with peptide AEEEIYGEFEAKKKG, and represented as above, and samples of the immunoprecipitates probed with anti-Flag Ab. n=2 per condition. p-values were calculated using an *Anova* test (Tukey's multiple comparisons test). (**B**) Western blot analyses of lysates from 293T cells expressing (left panel) Flag-tagged WT or mutant forms ACK1 probed with anti-ACK1 Tyr(P)[284] (PY284), anti-Flag and anti-tubulin antibodies (Ab), and expressing (right panel) Flag-tagged WT or mutant BRK probed with anti-BRK Tyr(P)[342] (PY342), anti-Flag and anti-tubulin antibodies. For BRK, 293T cells were starved overnight, and stimulated with 100 ng/ml EGF for 10 min. The lysate from WT BRK indicated as low was from cells transfected with one-tenth the amount of WT DNA. (**C**) Western blot analyses of lysates from 293T cells expressing ACK1-Flag or BRK-Flag treated with AIM100 or Cpd4f and probed with anti-ACK1 Tyr(P)284 (PY284) or anti-BRK Tyr(P)342 (PY342) and anti-Flag antibodies. (**D**) May-Grunwald-Giemsa staining of iPSC-derived macrophages from familial controls and ACK1 and BRK patients. Scale bar 10 μm, 100 X objective. Representative images from over 50 observed cells per line. (**E**) Immunoprecipitation (IP) kinase assay in patients' macrophages. (Left panel) ACK1 was immunoprecipitated from BRK[WT/G321R], ACK1[WT/K161Q] and ACK1[A156T/K161Q] iPSC-derived macrophages with anti-ACK1 Ab. The immunoprecipitated proteins were used in duplicate for in vitro kinase reactions with WASP synthetic peptide. Samples of the immunoprecipitates were also probed with anti-ACK1 Ab and anti-tubulin Ab. (Right panel) BRK was immunoprecipitated from ACK1[WT/K161Q], ACK1[A156T/K161Q], BRK[WT/G321R] and BRK[G257A/G321R] iPSCs-derived macrophages with anti-BRK Ab. The immunoprecipitated proteins were probed with anti-BRK Tyr(P)[342] (PY342) and anti-BRK antibodies.

The online version of this article includes the following source data and figure supplement(s) for figure 2:

**Source data 1.** *Figure 2A*: Immunoprecipitation (IP) kinase activity assay of WT and mutant ACK1 and BRK kinases in HEK293T cells.

**Source data 2.** *Figure 2E*: Immunoprecipitation (IP) kinase activity assay in patients' macrophages.

**Source data 3.** Uncropped and labeled gels for *Figure 2*.

**Source data 4.** Raw unedited gels for *Figure 2*.

**Figure supplement 1.** Generation and characterization of control and patient derived iPSCs and iPSC-macrophages.

**Figure supplement 2.** Homozygote mutant alleles reported in public database gnomAD with MAF >0.005 do not affect the kinase activity of ACK1.

**Figure supplement 2—source data 1.** Raw unedited gels for *Figure 2—figure supplement 2*.

**Figure supplement 2—source data 2.** Raw unedited gels for *Figure 2-figure supplement 2*.

in comparison to controls (*Figure 2E*). The peptide substrate specificity of BRK overlaps with those of Src family kinases (*Qiu and Miller, 2002*) present in the anti-BRK IPs. We therefore studied BRK kinase activity by IP-WB, which showed reduced BRK Tyr342 phosphorylation in BRK[G257A/G321R] macrophages in comparison to controls (*Figure 2E*). These data altogether indicate that the patients' ACK1 A156T and BRK G321R variants are kinase-dead alleles and K161Q and BRK G257A are severe hypomorphs.

ACK1 and BRK sequences are highly conserved in human populations, when compared with other species (*Figure 1C*). Apart from the alleles we report here, only 90 and 35 predicted loss-of-function

(LOF) alleles are reported in gnomAD for *ACK1* or *BRK* respectively, with MAFs ≤10⁻⁴, and none of them reported in homozygosity. Conversely, we examined the in vitro kinase activity of *ACK1* mutants with a MAF ≥0.005 and reported as homozygous in gnomAD (n=4). In vitro kinase activity of these variants was normal (*Figure 2—figure supplement 2A–C*). There was no *BRK* mutant with MAF ≥0.005 reported as homozygous in gnomAD. Thus, autosomal recessive deficiency for either protein, whether complete or partial, is expected to be well below 1/100,000 in the general population, which is compatible with autosomal recessive ACK1 and BRK deficiencies underlying SLE in patients from these two kindreds. To investigate whether *ACK1/TNK2* or *BRK/PTK6* were subject to selection, we gathered data using different metrics quantifying negative selection in the human genome. Analysis of f parameter from SnIPRE (*Eilertson et al., 2012*), lofTool (*Fadista et al., 2017*), evoTol (*Rackham et al., 2015*), and CoNeS (*Rapaport et al., 2021*) metrics, as well as intraspecies metrics from RVIS (*Petrovski et al., 2013*), LOEUF (*Karczewski et al., 2020*), and pLI / pRec (*Lek et al., 2016*) suggested that the genes are not under strong negative selection, consistent with the deficiency being recessive (*Figure 2—figure supplement 2D*).

## ACK1 and BRK inhibition promotes autoimmunity in Balb/c mice

ACK1- and BRK-deficient T cells are characterized by increased proliferation and activation (*Kasprzycka et al., 2006*; *Sridaran et al., 2022*). However, ACK1- and BRK-deficient mice do not develop Lupus-like disease on a C57BL/6 background (*Sridaran et al., 2022*; *Haegebarth et al., 2006*). We therefore investigated the consequences of inhibiting ACK1 and BRK kinase activity in wild-type (WT) BALB/cByJ female mice which are more susceptible to developing autoimmunity. For this purpose, groups of BALB/cByJ mice were either left untreated of received a single intraperitoneal injection of pristane to induce a lupus-like disease (*Satoh and Reeves, 1994*) and a weekly injection of ACK1 or BRK inhibitors AIM100 (*Mahajan et al., 2012*) and Cpd4f (*Oelze et al., 2015*), or DMSO vehicle alone for 12 weeks. In the absence of pristane, mice that received ACK1 or BRK inhibitors developed a large array of circulating anti-nuclear IgG antibodies, including but not limited to autoantibodies associated with SLE such as anti-histones, anti-chromatin, anti U1-snRNP, anti-SSA, and anti-Ku (*Figure 3A*). These data suggested that ACK1 and BRK inhibition are sufficient to promote autoimmunity in mice. We did not observe glomerular deposit of IgG at 12 weeks in these mice (*Figure 3B and C*, *Figure 3—figure supplement 1*). In contrast, BALB/cByJ mice which received pristane treatment (*Satoh and Reeves, 1994*) in addition to either ACK1 or BRK inhibitors had increased kidney glomerular deposits of IgG as well as increased serum autoantibody levels including anti-Ku (p70/p80), LA/SSB, Ro-SSA, Histone H4 and H2B, in comparison to DMSO vehicle controls (*Figure 3A–C*, *Figure 3—figure supplement 1*). Therefore, inhibition of ACK1 or BRK increase serum autoantibody titers and worsen pristane-induced Lupus in WT BALB/cByJ mice. Together with the above genetic analysis, these findings support the hypothesis that autosomal recessive ACK1 and BRK kinase deficiency may underlie or contribute to the development of SLE in children and young adults depending on genetic and environmental context.

## ACK1 and BRK kinase domain variants may lose the ability to link MERTK to RAC1, AKT and STAT3 activation for efferocytosis

NRTKs, including ACK1 and BRK, regulate phosphorylation of downstream effectors/ adaptor proteins involved in cell activation, migration, and proliferation including RAC1, AKT, STATs, and ERK (*Tsui and Miller, 2015*; *Kamalati et al., 2000*; *Zheng et al., 2010*; *Mahajan et al., 2010*; *Liu et al., 2006*; *Mahendrarajah et al., 2017*; *Mao and Finnemann, 2015*; *van der Horst et al., 2005*). NRTK deficiency can result in defective regulation of immune cell activation and survival which can lead to autoimmunity (*Kasprzycka et al., 2006*; *Sridaran et al., 2022*; *Yu et al., 2001*; *Chan et al., 1997*; *Hanke et al., 1996*; *Hibbs et al., 1995*; *Lamagna et al., 2014*; *Nishizumi et al., 1995*). We found that, in contrast to the reference ACK1 and BRK alleles, the patient's ACK1 and BRK variant alleles do not phosphorylate AKT and STAT3 (*Figure 4A and B*), and do not activate RAC1 to generate RAC-GTP (*Figure 4C*). NRTKs such as ACK1 (*Mahajan et al., 2005*) and PTK2/FAK (*Wu et al., 2005*) are also downstream targets of the TAM family receptor MERTK which is expressed on macrophages and controls the anti-inflammatory engulfment of apoptotic cells, a process known as efferocytosis (*Scott et al., 2001*; *Henson and Bratton, 2013*; *Henson, 2017*). Efferocytosis allows for the clearance of apoptotic cells before they undergo necrosis and release intracellular inflammatory molecules,

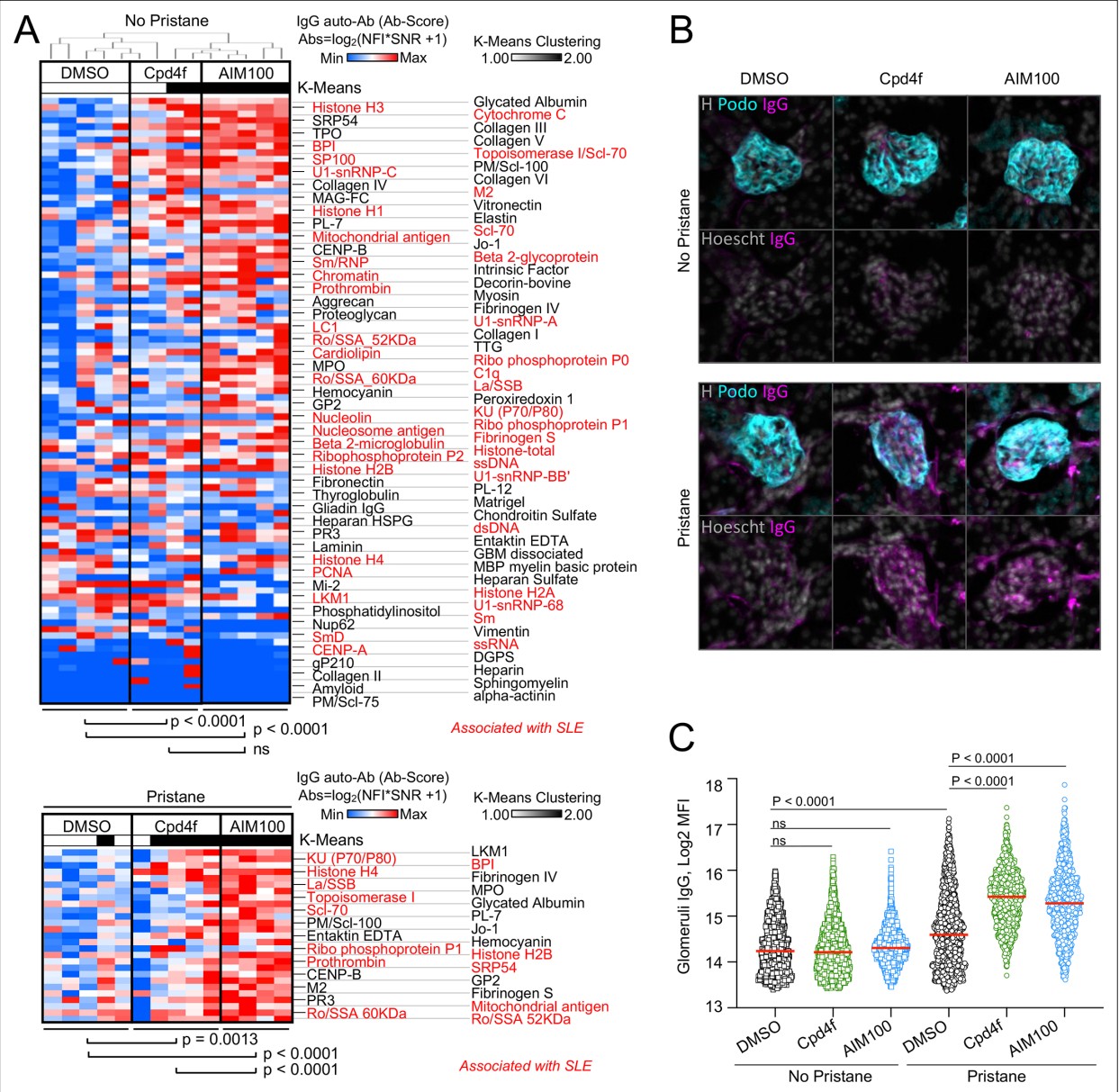

**Figure 3.** ACK1 and BRK blockade induces autoimmunity in mice. (**A**) Heatmaps comparing the levels of IgG autoantibodies detected in serum of mice treated with inhibitors. Heatmaps show autoantigen microarray panels performed on serum from 4-month-old BALB/cByJ female mice which received a weekly intra-peritoneal injection of DMSO (vehicle, 20 µl/mice), AIM100 (25 mg/kg in 20 µl), or Cpd4f (20 mg/kg in 20 µl) since the age of 5 weeks. *Top panel* depicts results for mice that did not receive a pristane injection. *Bottom panel* represents results for the top differentially produced auto-antibodies in inhibitor treated or control mice that received a Pristane injection at the age of 5 weeks. Plotted values represent Ab Scores (Log$_2$ [antigen net fluorescence intensity (NFI) x signal-to-noise ratio (SNR) +1]). Heatmap columns represent serum analysis of independent mice (n=4–5 for each of the 3 conditions). Heatmap rows sorted top to bottom starting with most significantly increased Ab Score in Cpd4f and AIM100 mice in comparison to DMSO-treated mice. p-Values were calculated using a Wilcoxon matched-pairs signed rank tests. Hierarchical clustering is based on one minus Pearson correlation with complete linkage method. K-means clustering is based on Euclidean distance, with two clusters, with 10,000 maximum iterations. (**B,C**) Immunofluorescence for mouse IgG on kidney sections. Representative micrographs (**B**) displaying glomeruli on kidney sections from 4-month-old BALB/cByJ female mice treated as in (**A**) and stained with Hoechst 33342, anti-mouse IgG, and anti-mouse Podoplanin antibody. In the quantification plot (**C**) each symbol represents the IgG mean fluorescence intensity (MFI) in a single glomerulus, of mice treated with designated inhibitors, in the presence or absence of pristane. Approximately 250 glomeruli we analyzed per section/mouse (>95% of all glomeruli in an entire longitudinal kidney section). n=4–5 mice per condition. p-Values were obtained using a Kruskal-Wallis test with multiple comparisons.

The online version of this article includes the following source data and figure supplement(s) for figure 3:

**Source data 1.** *Figure 3A*: Heatmaps comparing the levels of IgG autoantibodies detected in serum of control and inhibitor treated mice.

*Figure 3 continued on next page*

*Figure 3 continued*

**Source data 2.** *Figure 3C*: Quantification of glomerular IgG in kidney sections of control and inhibitor treated mice.

**Figure supplement 1.** Extended analysis of ACK1 and BRK inhibitor treated mice.

**Figure supplement 1—source data 1.** Related to *Figure 3—figure supplement 1A*: Heatmap comparing the levels of select IgG autoantibodies detected in serum of control and inhibitor treated mice.

and simultaneously leads to increased production of anti-inflammatory molecules (TGFβ, IL-10, and PGE2) and a decreased secretion of proinflammatory cytokines (TNF-alpha, IL-1β, IL-6; *Scott et al., 2001*; *Henson and Bratton, 2013*; *Henson, 2017*; *deCathelineau and Henson, 2003*; *Nagata, 2018*; *Fadok et al., 1998*; *Voll et al., 1997*; *Rothlin et al., 2015*; *Cvetanovic and Ucker, 2004*). In line with these findings, mice deficient in molecular components used by macrophages to efficiently perform efferocytosis, such as MFG-E8, MERTK, TIM4, and C1q, develop phenotypes associated with autoimmunity (*Scott et al., 2001*; *Cohen et al., 2002*; *Henson and Bratton, 2013*; *Nagata, 2018*; *Fadok et al., 1998*; *Voll et al., 1997*; *Cvetanovic and Ucker, 2004*; *Hanayama et al., 2004*; *Miyanishi et al., 2012*; *Colonna et al., 2016*; *Nagata et al., 2010*; *Kimani et al., 2014*; *Hanayama et al., 2002*; *Kawano and Nagata, 2018*; *Watanabe-Fukunaga et al., 1992*; *Singer et al., 1994*). Furthermore, defects in efferocytosis are also observed in patients with SLE and glomerulonephritis (*Nagata, 2018*; *Herrmann, 1998*; *Baumann, 2002*; *Schrijvers et al., 2005*; *Morioka et al., 2019*).

In IP kinase assays, MERTK activated the kinase activity of wild-type ACK1 (*Mahajan et al., 2005*) but not of the ACK1 A156T and ACK1 K161Q variant alleles (*Figure 4D*). In addition, MERTK also activated BRK kinase activity, but the BRK G321R and G257A alleles were kinase dead and hypomorph variants, respectively (*Figure 4D*). MERTK mediates recognition of PtdSer on apoptotic cells via GAS6 and Protein S (*Scott et al., 2001*; *Seitz et al., 2007*; *Cohen et al., 2002*) leading to their engulfment, which involves activation of RAC1 for actin reorganization and the formation of a phagocytic cup (*Wu et al., 2005*; *Mao and Finnemann, 2015*). PtdSer recognition also typically stimulates an anti-inflammatory process mediated in part via AKT (*Sen et al., 2007*) and STAT3 and their target genes such as *SOCS3* (*Yi et al., 2009*; *Vergadi et al., 2017*; *Byles et al., 2013*; *Liao et al., 2011*; *Roberts et al., 2017*; *Matsukawa et al., 2005*; *Sica and Mantovani, 2012*) and results in the inhibition of LPS-mediated production of inflammatory mediators such as TNF and IL1β, and the production of cytokines such as IL-10, TGFβ (*Henson and Bratton, 2013*; *Fadok et al., 1998*; *Voll et al., 1997*; *Rothlin et al., 2015*; *Cvetanovic and Ucker, 2004*). Altogether, these data raised the hypothesis that one of the consequences of the defective activity of ACK1 and BRK kinase variants might be an impaired efferocytic response to PtdSer on apoptotic cells.

## ACK1 and BRK kinase deficiency disrupts the anti-inflammatory response driven by apoptotic cells in macrophages

Efferocytosis can be carried out by multiple cell types; however, macrophages are the main contributors in this process (*Roberts et al., 2017*; *Parnaik et al., 2000*; *Nishi et al., 2014*; *Lazarov et al., 2023*). MERTK kinase activity mediates efferocytosis by human iPSC-derived macrophages (*Wanke et al., 2021*). We thus examined the transcriptional responses of controls, patients, and inhibitor-treated iPSC-derived macrophages to apoptotic thymocytes by RNA-seq (*Figure 4—figure supplement 1*). GSEA analysis of differentially expressed genes indicated that, in contrast to control, ACK1- and BRK-deficient macrophages, as well as WT macrophages treated with ACK1 or BRK inhibitors failed to upregulate gene sets associated with AKT signaling and the negative regulation of the inflammatory response (*Figure 4E*). Transcriptional repressors including the AKT targets *ATF3*, *TGIF1*, *NFIL3*, and *KLF4*, the STAT3 targets *SOCS3* and *DUSP5*, as well as *CEBPD* and the inhibitor of E-BOX DNA Binding *ID3* were among the top-ten genes which expression is induced by apoptotic cells in WT macrophages (*Figure 4F*), but this regulation was lost in mutant and inhibitor-treated macrophages (*Figure 4F*). ATF3, TGIF1, NFIL3, and KLF4 are involved in the negative regulation of inflammation in macrophages (*Vergadi et al., 2017*; *Byles et al., 2013*; *Liao et al., 2011*; *Roberts et al., 2017*), SOCS3 is an inhibitor of the macrophage inflammatory response and DUSP5 is a negative regulator of ERK activation (*Matsukawa et al., 2005*; *Sica and Mantovani, 2012*; *Seo et al., 2017*). These data suggest that the kinase domain of ACK1 and BRK contribute to the macrophage anti-inflammatory gene expression program driven by apoptotic cells.

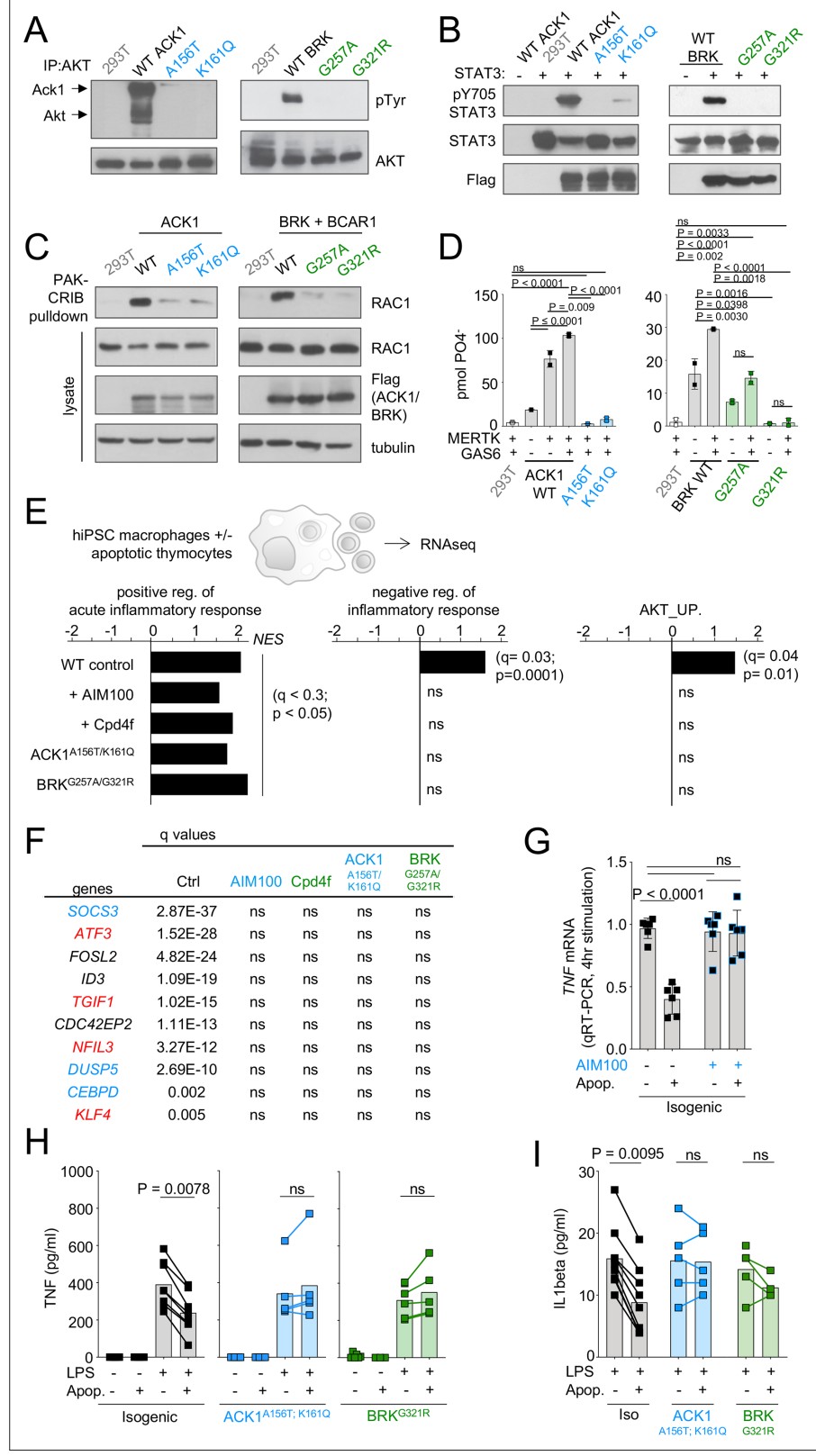

**Figure 4.** ACK1 and BRK kinase deficiency disrupts the anti-inflammatory response driven by apoptotic cells in macrophages. (**A**) Western blot analysis for AKT phosphorylation by ACK1 and BRK. Cell lysates from 293T cells were incubated with anti-AKT. Immunoprecipitated proteins were probed with anti-phosphotyrosine and anti-AKT antibodies. (**B**) Western blot analysis for STAT3 phosphorylation by ACK1 and BRK. Lysates from 293T

*Figure 4 continued on next page*

*Figure 4 continued*

cells coexpressing STAT3 and Flag-tagged WT or mutant forms (A156T and K161Q) of ACK1 or mutant forms (G257A and G321R) of BRK were probed with anti-phospho-STAT3 (Tyr705), anti-STAT3 and anti-Flag antibodies. For analysis of BRK, cells were treated with 100 ng/ml EGF for 10 min. (C) RAC activation by WT ACK1 and BRK. Cell lysates from 293T cells expressing WT or mutant forms of ACK1 (*left*) and lysates from 293T cells expressing WT or mutant forms of BRK (*right*) were incubated with GST-PAK CRIB sepharose beads, and the level of RAC1 GTP was determined by immunoblotting with anti-Rac1 antibody. Lysates were also probed with anti-Rac1, anti-FLAG and anti-tubulin antibody. For analysis of BRK, 293T cells were cotransfected with CAS and stimulated with 100 ng/ml EGF for 10 min. (D) MERTK increases kinase activity of BRK and ACK1. IP kinase assay. ACK1 (left) was immunoprecipitated from 293T cells co-transfected with Flag-tagged ACK1 WT, ACK1 A156T, or ACK1 K161Q and MERTK with anti-Flag Ab. Immunoprecipitated proteins were used in duplicate in vitro for kinase reactions with WASP synthetic peptide and results represented as pmol phosphate transferred. BRK (right) was immunoprecipitated as above from 293T cells co-transfected with Flag-tagged BRK WT or mutants and MERTK with anti-Flag Ab. Kinase reactions was performed with peptide AEEEIYGEFEAKKKG, and represented as above. p-Values were calculated using an Anova test (Tukey's multiple comparison test). (E) Regulation of inflammatory response. Significant normalized enrichment scores (NES) for GO 'positive regulation of acute inflammation' gene set, GO 'negative regulation of inflammatory response' gene set, and GO 'AKT_UP.V1_UP' gene set in WT and mutant macrophages, and WT treated with AIM100 (2 µM) or Cpd4f (0.5 µM), exposed to apoptotic cells, with three replicates per experimental condition. Significant enrichment (p-value <0.05 and FDR (q-value) <0.25) are calculated as reported in Materials and methods. (F) Table of the top 10 differentially regulated genes by apoptotic cells in WT macrophages are not differentially expressed in mutant macrophages and WT macrophages treated with AIM100 or Cpd4f (treated as in E). Numbers indicate FDR (q-value). Known target genes of STAT3 and AKT are labeled in blue and red respectively (G) *TNF* mRNA production by WT macrophages treated with AIM100 (2 µM) 4 hr after exposure to apoptotic cells. n=6, from two independent experiments. (H,I) TNF and IL1β production by macrophages, as measured by ELISA on media collected from mutant and isogenic WT macrophages (C12.1) incubated with mouse apoptotic thymocytes for 90 min, then stimulated with LPS (1 ng/ml) for 18 hr. n≥4, from ≥2 independent experiments. p-Values in H were calculated by Wilcoxon matched-pairs signed rank tests for data that is not normally distributed, while p-values in G and I were calculated using an *Anova* test with Tukey's correction for multiple comparisons.

The online version of this article includes the following source data and figure supplement(s) for figure 4:

**Source data 1.** *Figure 4D*: Immunoprecipitation (IP) kinase activity assay of WT and mutant ACK1 and BRK kinases in HEK293T cells in the presence of MERTK and/or GAS6.

**Source data 2.** *Figure 4E*: GSEA of control, inhibitor treated, and mutant iPSC-derived macrophages in the presence or absence of apoptotic cells.

**Source data 3.** *Figure 4G*: TNF mRNA production by WT AIM100 treated macrophages 4 hr after exposure to apoptotic cells.

**Source data 4.** *Figure 4H*: TNF protein production by WT and mutant iPSC-derived macrophages in the presence or absence of LPS and/or apoptotic cells.

**Source data 5.** *Figure 4I*: IL1b protein production by WT and mutant iPSC-derived macrophages in the presence of LPS and/or apoptotic cells.

**Source data 6.** Uncropped and labeled gels for *Figure 4*.

**Source data 7.** Raw unedited gels for *Figure 4*.

**Figure supplement 1.** Principal component analysis (PCA) and differentially expressed genes in RNA sequencing datasets.

**Figure supplement 2.** Generation and characterization of isogenic control and mutant iPSCs and iPSC-macrophages.

Decreased *TNF* gene expression by macrophages in response to apoptotic cells was prevented by the ACK1 inhibitor (*Figure 4G*), but the production of TNF at the protein level is not detectable by ELISA in this model (see *Figure 4H*). MERTK-deficient mice are susceptible to LPS-induced endotoxic shock (*Camenisch et al., 1999*), and MERTK-dependent anti-inflammatory program elicited by apoptotic cells on macrophages is best evidenced by the reduction of LPS-mediated production of inflammatory mediators such as TNF or IL1β (*Fadok et al., 1998*; *Voll et al., 1997*; *Cvetanovic and Ucker, 2004*; *Sen et al., 2007*; *Camenisch et al., 1999*). We thus tested the decrease of LPS-induced production of TNF and IL1β by apoptotic cells. For this purpose, we generated isogenic variants and control hiPSCs and hiPSCs-derived macrophages from the same donor (*Figure 4—figure supplement 2*).

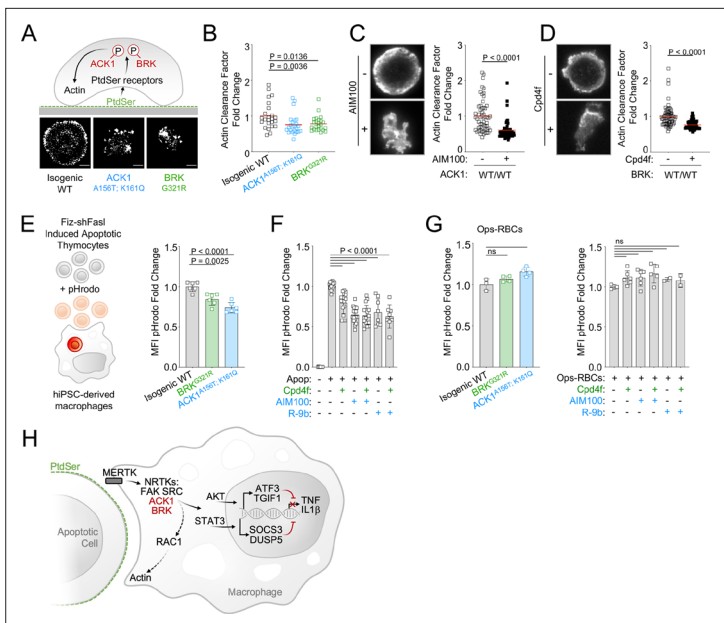

**Figure 5.** ACK1 and BRK kinase deficiency alter actin remodeling at the phagocytic cup and modestly decrease engulfment of apoptotic cells in macrophages. (**A**) Actin remodeling in macrophages. Schematic and representative images of of F-actin by TIRF microscopy in macrophages of indicated genotype, deposited on PtdSer-coated plates for 20 min. (**B**) Quantification of actin clearance factor for macrophages of the indicated genotypes. Actin remodeling (actin clearance factor) was calculated as a ratio of F-actin staining intensity at cell border divided by F-actin staining intensity at cell center. The actin clearance factor ratios were normalized to the mean value of WT control. Each replicate indicates actin clearance factor fold change from WT mean in single cells. n>20, from two independent experiments. Red lines denote the mean. (**C, D**) Actin remodeling quantification (as in A,B) and representative TIRF images of WT macrophages (C12.1 line) pretreated with DMSO, AIM100 (2 µM) or Cpd4f (0.5 µM). n>24, from three independent experiments. p-Values in B-D were obtained using a Mann-Whitney test. (**E, F**) Uptake of apoptotic cells. (**E**) Schematic depicts uptake of apoptotic mouse thymocytes treated with Fiz-shFASL and labeled with the pH-sensitive dye pHrodo by iPSC-derived macrophages. Isogenic WT (C12.1 line) and isogenic ACK and BRK point mutant macrophages were incubated with pHrodo-labeled mouse apoptotic thymocytes for 90 min and analyzed by flow cytometry. Graph represents mean pHrodo fluorescence Intensity (MFI) fold change calculated by dividing total pHrodo MFI (610/20 nm) of individual samples by the average MFI of isogenic WT macrophages. n≥3, from three independent experiments. p-Values were obtained using an *Anova* test with Tukey's correction for multiple comparisons. (**F**) Uptake of apoptotic cells as in (**E**) with WT macrophages (C12.1 line) pretreated with AIM100 (2 µM), R-9b (4 µM), Cpd4f (0.5 µM), or DMSO. n≥8, from ≥4 independent experiments. (**G**) Uptake of opsonized sheep red blood cells. WT macrophages (C12.1 line) are pretreated as in (**F**) and incubated with opsonized pHrodo sheep red blood cells for 90 min. Graphs represent mean fluorescence Intensity (MFI) fold change calculated by dividing total pHrodo MFI (610/20 nm) of individual samples by the average MFI of WT macrophages. n≥2, from two independent experiments. p-Values are obtained using an *Anova* test with Tukey's correction for multiple comparisons. (**H**) Schematic representation of ACK1 and BRK proposed function in efferocytosis.

The online version of this article includes the following source data and figure supplement(s) for figure 5:

**Source data 1.** *Figure 5B*: Frustrated engulfment assays for actin remodeling quantification in response to PtdSer in WT and mutant iPSC-derived macrophages.

**Source data 2.** *Figure 5C*: Frustrated engulfment assay for actin remodeling quantification in response to PtdSer in AIM100 treated WT macrophages.

**Source data 3.** *Figure 5D*: Frustrated engulfment assay for actin remodeling quantification in response to PtdSer in Cpd4f treated WT macrophages.

**Source data 4.** *Figure 5E*: Quantification of apoptotic cell uptake by WT and mutant iPSC-derived macrophages.

**Source data 5.** *Figure 5F*: Quantification of apoptotic cell uptake by inhibitor-treated and control WT iPSC-derived macrophages.

**Source data 6.** *Figure 5G*: Quantification of opsonized red blood cell uptake by inhibitor-treated and control WT and mutant iPSC-derived macrophages.

*Figure 5 continued on next page*

*Figure 5 continued*

**Figure supplement 1.** Engulfment of beads, *Escherichia coli*, and *Candida albicans*.

Both isogenic variants and control macrophages produced similar amounts of TNF in response to LPS (*Figure 4H*). However, exposure to apoptotic cells inhibited TNF production in control macrophages by 50% but did not inhibit TNF production in ACK1 and BRK-deficient macrophages (*Figure 4H*). Similarly, isogenic variants and control macrophages produced similar amounts of IL1β in response to LPS (*Figure 4I*), but apoptotic cells only decreased IL1β production by isogenic macrophages and not by mutant macrophages (*Figure 4I*). These data altogether indicate that ACK1 and BRK kinase activities contribute to the macrophage anti-inflammatory response during efferocytosis and are required for the decrease of TNF and IL1β production induced by LPS in response to apoptotic cells, a hallmark of their anti-inflammatory effect on macrophages (*Henson and Bratton, 2013*; *Fadok et al., 1998*; *Voll et al., 1997*; *Cvetanovic and Ucker, 2004*).

## ACK1 and BRK kinase deficiency alter actin remodeling at the phagocytic cup and modestly decrease engulfment of apoptotic cells in macrophages

MERTK-dependent signaling for anti-inflammatory response and for cargo engulfment driven by recognition of PtdSer on apoptotic cells are distinct and separable molecular events (*Tibrewal et al., 2008*; *Elliott and Ravichandran, 2010*), however because RAC1, which controls engulfment of apoptotic cells (*Albert et al., 2000*), is a target of ACK1 and BRK (see *Figure 4C*) as well as PTK2/FAK (*Wu et al., 2005*), we investigated the engulfment of apoptotic cells by ACK1- and BRK-deficient macrophages. We assessed actin-ring formation by total internal reflection fluorescence (TIRF) microscopy in frustrated engulfment assays on PtdSer-coated glass slides. Wild-type iPSC-derived macrophages formed a typical actin-ring; however, isogenic *ACK1* and *BRK* mutant macrophages presented with an altered actin-ring (*Figure 5A*), and a reduced actin clearance factor (*Figure 5B*), indicating an impairment of actin remodeling following binding to PtdSer. The actin-ring was also altered and actin clearance factor was decreased in WT macrophages treated with ACK1 and BRK inhibitors in comparison to controls (*Figure 5C and D*). These data suggest the kinase activity of ACK1 and BRK contributes to link PtdSer receptors to cytoskeleton rearrangement at the phagocytic synapse.

We therefore investigated whether this defect translated to a reduced uptake of apoptotic cells by macrophages. We found that engulfment of apoptotic thymocytes labeled with the pH-sensitive probe pHrodo (*Nishi et al., 2014*; *Miksa et al., 2009*; *Shiraishi et al., 2004*) was only moderately reduced, by 10–20% in *ACK1* and *BRK* isogenic mutant macrophages (*Figure 5E*). This phenotype, although modest, was reproducible by a 30-min exposure of WT macrophages to two different ACK1 inhibitors, AIM100 (2 µM) and R-9b (*Lawrence et al., 2015*; 4µM), and to the BRK inhibitor Cpd4f (0.5 µM; *Figure 5F*). This reduced uptake of apoptotic cells was not attributable to a global engulfment defect, because ACK1 and BRK genetic kinase deficiency or inhibitors did not prevent Fc-dependent phagocytosis of a large cargo such as opsonized red blood cells (*Figure 5G*), or the uptake of polystyrene beads (*Figure 5—figure supplement 1A*) or microorganisms such as bacteria and fungi (*Figure 5—figure supplement 1B, C*). Altogether, these data show that the kinase activity of ACK1 and BRK participate to the formation of phagocytic synapse but are largely dispensable for engulfment of apoptotic cells and are not required for phagocytosis of microbes and opsonized cargo.

## Discussion

In this study, we combined whole exome sequencing and forward genetic analysis in multiplex SLE families, with a biochemical analysis of genetic variants, murine studies, and functional approaches in human iPSC-derived macrophages to identify novel genes the mutations of which may underlie SLE. In two unrelated families, we identified compound heterozygous loss-of-function or hypomorph variants in the kinase domains of two non-receptors tyrosine kinases, *TNK2/ACK1* and *PTK6/BRK*. Patients from the two families were children or young adults and presented with a severe clinical form of SLE, lupus nephritis. However, we analyzed 27 GWAS studies of SLE (https://www.gwascentral.org/), and none of them reported a common variant within close genetic distance of the *TNK2/ACK1* or *PTK6/*

*BRK* genes with a p-value lower than $5 \times 10^{-8}$, a statistical threshold of genome-wide significance for GWAS. ACK1 and BRK deficiency are thus likely to only account for the genetic basis of SLE in a minority of patients. Nevertheless, ACK1 and BRK kinase inhibitors aggravate autoimmunity and IgG glomerular deposits in BALB/cByJ mice. Altogether, the present data indicate that autosomal recessive TNK2/ACK1 and PTK6/BRK kinase deficiency probably underlie the development of SLE in a small proportion of children and young adults, depending on genetic and environmental context.

Defective efferocytosis has been shown to contribute to autoimmunity in mice and is relevant to the pathogenesis of SLE (*Nagata, 2018*; *Hanayama et al., 2004*; *Miyanishi et al., 2012*; *Nagata et al., 2010*; *Kimani et al., 2014*; *Hanayama et al., 2002*; *Kawano and Nagata, 2018*; *Herrmann, 1998*; *Baumann, 2002*). Our results also suggest that TNK2/ACK1 and PTK6/BRK kinase deficiencies, in addition to dysregulating B and T cell survival and activation (*Kasprzycka et al., 2006*; *Sridaran et al., 2022*), also impair the MERTK-dependent anti-inflammatory response of the patients' macrophages to apoptotic cells during efferocytosis, and to a lesser extent the engulfment of the apoptotic cells. MERTK-dependent engulfment of apoptotic cells and anti-inflammatory response were shown to be distinct and separable (*Tibrewal et al., 2008*; *Elliott and Ravichandran, 2010*). The NRTK PTK2/FAK and Src are important for the former (*Tibrewal et al., 2008*; *Tao et al., 2015*), while PTK2/FAK is dispensable for the latter (*Tibrewal et al., 2008*). Our experiments suggest that in contrast, TNK2/ACK1 and PTK6/BRK are more important for the control of TNF and IL1β production than for engulfment itself. Altogether, our observations identify a rare Mendelian cause of severe SLE and a role for the NRTK TNK2/ACK1 and PTK6/BRK in efferocytosis, thereby contributing to a molecular and cellular dissection of SLE.

# Materials and methods
## Human sample collection and consent information
The study was approved by the Institutional Review Board of St Thomas' Hospital; Guy's hospital; the King's College London University and the Memorial Sloan Kettering Cancer Center. All subject samples were obtained after written informed consent from patients and their families according to the Helsinki convention (Ethics approval: 11/LO/1433). Ten multiplex families with lupus have been enrolled from Guy's and St Thomas' NHS Foundation Trust and UCL Hospital in London, UK from July 2010 to January 2012, following stringent criteria: a severe phenotype (lupus nephritis for at least one patient in each family), and a familial disease (≥2 family members affected in first degree). A total of 24 patients and 17 healthy controls from different ethnic origins (5 African ancestry, 4 Asian ancestry and 1 European ancestry) were selected. The patients each met the Systemic Lupus International Collaborating Clinics (SLICC) classification criteria for SLE (*Petri et al., 2012*). Lupus nephritis was confirmed by a kidney biopsy classified per the 2004 ISN/RPS (International Society of Nephrology/ Renal Pathology Society) classification and verified independently by two renal histopathologists.

One hundred Mauritian participants were enrolled under the Ethical Clearance provided by the University of Mauritius Research Ethics Committee. Written consent with due signatures was recorded from all participants prior to partaking in the study. Consent was documented on a confidential form in duplicate, with one copy given to the participants for their records. The University of Mauritius Research Ethics Committee approved, sanctioned and fully endorsed this mode of consent recording. The ethnic backgrounds of the 100 Mauritian participants consisted of 26 Creole, 16 Franco-Mauritian, 21 Indo-Mauritian, 2 Sino-Mauritian, 24 other or undisclosed Mauritians.

## Genetic analysis
Whole Exome Sequencing (WES) of the 10 multiplex families (patients and familial healthy controls) was performed at the New York Genomics Centre on an Illumina HiSeq 2000 sequencing machine. Genomic DNA extracted from the patients and familial healthy control's peripheral blood cells were sheared with a Covaris S2 Ultrasonicator. An adapter-ligated library was prepared with the Paired-End Sample Prep kit V1 (Illumina). Exome capture was performed with the SureSelect Human All Exon kit (Agilent Technologies). Paired-end sequencing was performed on a HiSeq 2000, generating 100-base reads. For sequence alignment, variant calling and annotation, we used BWA aligner (*Li and Durbin, 2009*) to align sequences with the human genome reference sequence (hg19 build). Downstream processing was performed with the Genome analysis toolkit (*McKenna et al., 2010*), SAMtools (*Li*

*et al., 2009*), and Picard Tools. Substitution and indel calls were identified with a GATK Unified Genotyper and a GATK Indel GenotyperV2, respectively. All calls with a read coverage ≤2 x and a Phred-scaled SNP quality of ≤20 were filtered out. All the variants were annotated with the GATK Genomic Annotator. Variants were annotated following their minor allele frequency (MAF) in Exome Variant Server, 1000 Genomes Project, and Genome Aggregation Database (gnomAD), including the MAF in each ethnic subpopulation from gnomAD.

First, the variants have been prioritized at the gene level. We used the gene damage index (*GDI*) which provides the accumulated mutational damage of each human gene in healthy human population, based on the 1000 Genomes Project database (Phase 3) gene variations of healthy individuals and of the CADD score for calculating impact (*Itan et al., 2015*). GDI is very effective to filter out variants harbored in highly damaged (high GDI) genes that are unlikely to be disease-causing. We used a cut-off of 13.84, the recommended GDI value above which a gene is unlikely to be disease-causing which is the 95% CI upper boundary value, removing 5% of the genes in our analysis.

Secondly, we prioritized variants at the allele level. We excluded variants that were too frequent in our in-house database to explain the disease (*Maffucci et al., 2019*). We then filtered out variants based on the predicted damaging impacts. The deleteriousness of each variant was assessed using in silico algorithms: CADD (http://cadd.gs.washington.edu/score) and Mutation Significance Cutoff browser (MSC, http://pec630.rockefeller.edu/MSC/). MSC scores were generated using 99% confidence interval based on the CADD 1.3 scores of all disease causing-mutations in Human Gene Mutation Database for any given gene (*Kircher et al., 2014*; *Itan et al., 2016*). We kept only the variants with a CADD/MSC higher than 1.

We thus analyzed our WES with the remaining variants by keeping all non-synonymous coding variants and essential splice site variants with a MAF according to the genetic model: (1) heterozygous variations with MAF $<10^{-4}$ in all ethnic subpopulations under an autosomal dominant (AD) model; (2) homozygous or compound heterozygous variants with MAF $<10^{-2}$ in all ethnic subpopulations under an autosomal recessive (AR) model; and (3) hemizygous (male) or homozygous (female) variations with MAF $<10^{-4}$ in all ethnic subpopulations under X-linked genetic models.

## Principal component analysis

PCA was performed with 2504 individuals of the 1000 Genomes database using WES high-quality variants as described in Belkadi et al. We searched for the closest neighbors of the patients in terms of ethnic origin using an Euclidian distance computed from the 10 first PCs (*Belkadi et al., 2016*; *Mullaert et al., 2021*).

## Biochemical analysis

### Cloning and Site-directed mutagenesis

The expression vectors for Flag- and HA-tagged Ack1 and for Flag-tagged Brk were described previously (*Prieto-Echagüe et al., 2010b*; *Prieto-Echagüe et al., 2010a*; *Qiu and Miller, 2004*). Site-directed mutagenesis was performed using the QuikChange Kit (Agilent; 200523). The expression vector for MerTK, pIRES2-EGFP Mer, was a gift from Dr. Raymond Birge (*Wu et al., 2005*).

### Cell transfection, immunoprecipitation, and western blotting

Cells were transfected 24 hr after plating with 8 µL polyethylenimine per µg of DNA in 150 mM NaCl. Cells were harvested 48 hr after transfection using lysis buffer (25 mM Tris, pH 7.5, 1 mM EDTA, 100 mM NaCl, 1% NP-40) supplemented with aprotinin, leupeptin, PMSF, and Na₃VO₄. For Western blotting, lysates were resolved by SDS-PAGE transferred to PVDF membranes, and probed with the appropriate antibodies. Horseradish peroxidase-conjugated secondary antibodies (GE Healthcare; NA931V; NA9340V) and western blotting substrate (Thermo Fisher Scientific; 32106) were used for detection.

For immunoprecipitation studies, cell lysates (1 mg total protein) were incubated with 1–2 µg of the appropriate antibody and 25 µL of protein A agarose (Roche; 11134515001) for at 4 °C for 4 hr-overnight. Anti-Flag immunoprecipitations were done with anti-Flag M2 affinity resin (Sigma). The beads were washed three times with lysis buffer, then eluted with SDS-PAGE sample buffer and resolved by SDS-PAGE. The proteins were transferred to PVDF membrane for Western blot analysis.

## Immunoprecipitation kinase assay

IP-kinase assays were carried out essentially as previously described (*Tsui and Miller, 2015*). Cell lysates (1 mg protein) were incubated with 25 µL of anti-Flag M2 affinity resin on a rotator for 4 °C for 4 hr-overnight, then washed three times with Tris-buffered saline (TBS). A portion of each sample was eluted with SDS-PAGE sample buffer and analyzed by anti-Flag western blotting. The remaining sample was used for a radioactive kinase assay. A WASP-derived peptide (sequence: KVIYDFIEKKKG; *Yokoyama et al., 2005*) was used as a substrate for Ack1, and a Src-specific peptide (sequence: AEEEEIYGEFEAKKKKG; *Qiu and Miller, 2002*; *Songyang et al., 1995*) was used as a substrate for Brk. The immunoprecipitated proteins were incubated with 25 µL of reaction buffer (30 mM Tris, pH 7.5, 20 mM MgCl$_2$, 1 mg/mL BSA, 400 µM ATP), 1 mM peptide, and 50–100 cpm/pmol of [γ-32P] ATP at 30 °C for 20 min. The reactions were terminated using 45 µL of 10% trichloroacetic acid. The samples were centrifuged and 30 µL of the reaction mixture was spotted onto Whatman P81 cellulose phosphate paper. After washing with 0.5% phosphoric acid, incorporation of radioactive phosphate into the peptide was measured by scintillation counting.

## Rac binding assay

The Cdc42/Rac interactive binding (CRIB) domain from PAK binds specifically to Rac in the GTP-bound state (*Brugnera et al., 2002*). The PAK-CRIB domain was expressed as a GST fusion protein in *E. coli* and purified with glutathione-agarose. The immobilized CRIB domain was incubated with cell lysates (1 mg total protein) for 2 hr at 4 °C. The resin was washed with TBS, and bound proteins were eluted with SDS-PAGE sample buffer and analyzed by anti-Rac western blotting.

# Cells

## HEK 293T

HEK 293T cells were maintained in Dulbecco's modified Eagle's medium (DMEM, Mediatech, Inc) supplemented with 10% fetal bovine serum (FBS; Sigma) and 1000 IU/ml penicillin and streptomycin.

## Derivation of human iPSCs

Generation of iPSCs from frozen peripheral blood mononuclear cells (PBMCs) was performed using a previously published protocol (*Yang, 2008*). Briefly, PBMCs were cultured in QBSF-60 media supplemented with L-Asorbic Acid (50 µg/mL), human SCF (50 ng/mL; R&D; 255-SC-010/CF), human IL-3 (10 ng/mL; Peprotech; 200–03), human EPO (2 U/mL; R&D; 287-TC-500), IGF-1 (40 ng/mL; R&D; 291-G1-200), and Dexamethasone (1 µM; Sigma; D8893-1MG) for 9–12 days to expand the erythroblast population. Then 4 Sendai viral vectors (Thermo Fisher Scientific; A16517) expressing Oct3/4, Sox2, Klf4, or c-Myc are used for transduction of 2.5x10$^5$ cells with 10 MOI for each virus for 24 hr. At day 2 post transduction, cells were plated in a 6 well gelatin coated plate containing MEFs (Thermo Fisher Scientific; A34181). After 9–12 days, small iPSCs colonies appear. At day 17–21, several colonies were picked and expanded individually into 1 well (12-well or 24-well plate) containing MEFs on gelatin in ESC media as detailed above, supplemented with 10 ng/ml basic fibroblast growth factor (bFGF; Peprotech; 100-18B). Five clones were established per cell line and were maintained in culture for 10 passages (2–3 months) to ensure stability of the lines. Two clones per cell line were selected and tested for chromosomal abnormality and showed a normal karyotype (46, XY or 46, XX). The other clones were frozen down.

## Culture of human iPSCs

Human iPSCs were maintained on mouse embryonic fibroblasts (MEFs, Thermo Fisher Scientific; A34181) in ESC media knock-out Dulbecco's modified Eagle medium (KO-DMEM, Thermo Fisher Scientific; 10829–018) with 20% KO serum replacement (Thermo Fisher Scientific; 10828–028), 2 mM L-glutamine (Thermo Fisher Scientific; 25030–024), 1% nonessential amino acids (Thermo Fisher Scientific; 11140–035), 1% penicillin/streptomycin (Thermo Fisher Scientific; 15140–122), 0.2% β-mer-captoethanol (Thermo Fisher Scientific; 31350–010) supplemented with 10 ng/ml basic fibroblast growth factor (bFGF, Peprotech; 100-18B). Passaging was performed every 7 days at 1:3-1:6 dilution ratio depending on the colony size. During passaging, iPSCs are detached as clusters by a 13min incubation at 37 °C with collagenase type IV (250 UI/ml final concentration; Thermo Fisher Scientific;

**Table 1.** sgRNA-target and ssODN sequences for generation of the TNK2 (ACK1) and PTK6 (BRK) isogenic mutant iPSC lines.

|  | sgRNA target | ssODN |
|---|---|---|
| TNK2 (A156T/K161Q) | GGCTCAGGACATCGGGCTTC | GCAGGTTGGCTCCGCGTGTCTGGGACCTGGCAAGTCCTGAGT CCTTGCAAATCCCGCTCTGGGCAGGTGAGTGTGACTGTGAAG TGCCTGCAGCCCGATGTCCTGAGCCAGCCAGAAGCCATGGAC GACTTCATCCGGGAGGTCAA |
| PTK6 (G321R) | CGCCAGGAACATCCTCGTCG | GCTGAGGGCATGTGTTACCTGGAGTCGCAGAATTACATCCAC CGGGACCTGGCCGCCAGGAACATCCTCGTCAGGGAAAACA CCCTCTGCAAAGTTGGGGACTTCGGGGTTAGCCAGGCTTATC AAGGTAGGGCCCTCAGAGGG |

17104019) and are pelleted at room temperature (RT) by centrifugation at 100 × *g*. The iPSC clusters are resuspended in ESC medium supplemented with 10 ng/ml bFGF (Peprotech; 100-18B) and plated on NUNC plates containing 12,500–16,000 MEFs per cm$^2$.

## Generation of the TNK2 (ACK1) and PTK6 (BRK) isogenic mutant iPSC lines

CRIPSR and single-stranded donor oligonucleotides (ssODN) were used as the tool to introduce the SNP mutations in hiPSCs. CRISPR sgRNA target was designed using the web resource at https:// www.benchling.com/crispr/. The target sequence was cloned into the pX330-U6-Chimeric_BB-CBh-hSpCas9 vector (Addgene plasmitd #42230) to make the gene targeting construct. The template ssODN was designed to carry the mutant nucleotide and served as the donor template. The ssODN was then purchased from IDT. The sgRNA-target and ssODN sequence are listed in *Table 1*.

To introduce the SNP mutations, WT iPSCs (C12) were dissociated using Accutase (Innovative Cell Technologies) and electroporated (1x10$^6$ cells per reaction) with 4 µg sgRNA-construct plasmid and 4 µl ssODN (10 µM stock) using Human Stem Cell Nucleofector solution (Lonza) following manufacturer's instructions. The cells were then seeded, and 4 days later, hESCs were dissociated into single cells by Accutase and re-plated at a low density (4 per well in 96-well plates) to get the single-cell clones. 10 days later, individual colonies were picked, expanded and analyzed by PCR and DNA sequencing. The PCR and sequencing primers are listed in *Table 2*.

## Differentiation of iPSC-derived macrophages

The hiPSCs to macrophage differentiation method was adapted from a previously published protocol (*Lachmann et al., 2015*). Briefly, newly passaged iPSCs were maintained from day 0 to day 3 in ESC media (see Culture of Human iPSCs) with 10 ng/ml bFGF and from day 3 to day 7 in ESC media without bFGF. At day 7, iPSCs colonies were detached in clusters using collagenase type IV (250 UI/ ml final concentration) (Thermo Fisher Scientific; 17104019) and transferred to six-well suspension plates in ESC media supplemented with 10 µM ROCK Inhibitor (Sigma; Y0503), on an orbital shaker at 100 rpm. The cell clusters were cultivated under these conditions from day 7 to day 13 to induce embryoid body (EB) formation. At day 13, well formed 200–500 µm EBs were manually picked under a microscope and transferred onto adherent tissue culture plates (~2.5 EBs/cm$^2$) for cultivation in APEL 2 medium (Stem Cell Tech; 05270) supplemented with 5% protein free hybridoma (Thermo Fisher Scientific; 12040077), 100 IU/ml penicillin and 100 µg/ml streptomycin (Thermo Fisher Scientific; 15140–122), 25 ng/ml human IL-3 (Peprotech; 200–03) and 50 ng/ml human M-CSF (Peprotech, 300–25). Starting from day 25 of the differentiation and then every week onwards for up to 5 weeks, suspension cells around EBs were carefully collected, filtered through a 100 µm mesh, plated at a density of ~15,000 cells/cm$^2$, and cultivated for 6–10 days in RPMI1640/GlutaMax (Thermo Fisher Scientific; 61870036) medium supplemented with 10% FBS (EMD Milipore TMS-013-B), and 100 ng/ ml human M-CSF and used further for functional analysis. All cells were cultured at 37 °C 5% CO$_2$ in standard tissue culture incubators.

**Table 2.** TNK2 (ACK1) and PTK6 (BRK) PCR and sequencing primers.

|  | PCR-Forward primer (used for sequencing) | PCR-Reverse primer |
|---|---|---|
| TNK2 | TGCTTACCCACCCAGATGAG | AAATCCAGAGACAGACCCGG |
| PTK6 | GAGAAAGTCCTGCCCGTTTC | GATTGCAGGTGTGTGGGGA |

## Mice

C57BL/6 J mice were used for preparation of apoptotic thymocytes and BALB/cByJ female mice were used for in vivo inhibitor treatment experiments (see below). Mice were purchased from The Jackson Laboratory. All mouse studies were performed in adherence with Institutional Review Board (IACUC 15-04-006) from MSKCC.

## Antibodies and flow cytometry

The following antibodies were used for human iPSCs-derived macrophages phenotyping: PE/Cy7 anti-human MERTK Ab (Biolegend; 367609); PE/Cy7 anti-human CD11b Ab (BioLegend; 301321); PE anti-human Intα5β3 Ab (R&D; FAB3050P); APC anti-human TIM4 Ab (BioLegend; 354007); AF647 anti-human TIM4 Ab (BioLegend; 354007); APC/Cy7 anti-human CD36 Ab (Biolegend; 336213); BV786 anti-human CD115 Ab (CSF-1R) (BD Biosciences; 743145); PE-Cy5 anti-human CD11c Ab (BD Biosciences; 561692); Alexa Fluor 700 anti-human HLA-DR Ab (BD Biosciences; 560743); APC/Cy7 anti-human CD45 Ab (BioLegend; 304014); BV650 anti-human CD14 Ab (BioLegend; 301836); Alexa Fluor 647 anti-human CD369 (Clec7A) Ab (BD Biosciences; 564855); Alexa Fluor 488 anti-human CD206 (MRC1) Ab (ThermoFisher Scientific; 564855).

iPSCs-derived macrophages were detached using trypsin (TrypLE Express, ThermoFisher Scientific; 12605–010), pelleted at 400 × $g$ for 5 min and resuspended in fluorescence-activated cell sorting (FACS) buffer (PBS +0.5% bovine serum albumin (BSA) +1 mM EDTA). After blocking Fc receptors (Miltenyi; 130-059-901) at 1/10 dilution for 10 min, the cells were washed with FACS buffer, pelleted at 400 G for 5 min, and immunostained in FACS buffer +antibody (1:50 to 1:200 dilution) for 30 min at 4 °C. Data were acquired on an ARIA III BD flow cytometer or a FACS Fortessa SORP instrument and analyzed with FlowJo. Dead cells and debris were excluded from the analysis using DAPI (1 µg/ml), side (SSC-A) and forward scatter (FSC-A) gating, and doublet exclusion using forward scatter width (FSC-W) against FSC-A. At least 5000 cells were acquired for each condition.

## Cytology

Cells were collected into FBS and centrifuged (800 rpm, 8 min, low acceleration) onto Superfrost slides (Thermo Fisher Scientific) using a Cytospin 3 (Thermo Shandon). Slides were air-dried for at least 30 min, and fixed for 5 min in methanol, stained in 50% May-Grunwald solution (Sigma-Aldrich, MG500) for 15 min, 5% Giemsa (Sigma-Aldrich, 48900) for 15 min, washed with Sorensons buffered distilled water (pH 6.8) for 5 min and rinsed with Sorensons buffered distilled water (pH 6.8). Slides were air-dried and mounted with Entellan New (Merck) and representative pictures were taken using an Axio Lab.A1 microscope (Zeiss) under a N-Achroplan 100 x/01.25 objective.

## Engulfment of eads, *E. coli*, or *C. albicans* by iPSC-macrophages

WT iPSCs-derived macrophages were plated at a density of 35,000 cells per well in 24-well plates and maintained in 0.5 ml of RPMI containing 10% FBS. 24 hr after plating, the macrophages were pretreated for 30 min with AIM100 (Tocris, 4946; 2 µM) and/or Cpd4f (Thermo Fisher Scientific, 53-100-00001; 0.5 µM) in RPMI without FBS prior to incubation with beads, *E. coli* or *C. albicans*.

Red fluorescent 2 µM Beads (Microparticles Invitrogen, F8826) were resuspended in RPMI ($5 \times 10^5$ beads/ml), sonicated for 5 min, and part of the suspension was warmed to 37 °C or cooled to 4 °C. 400 µl of the suspension was added per well ($2 \times 10^5$ beads/well) to pretreated iPSC-macrophages for each condition, and the plates were incubated for 1 hr at 37 °C or 4 °C.

pHrodo *E. coli* BioParticles (Invitrogen, P35361) were resuspended in RPMI (0.5 mg/ml), sonicated for 5 min, and part of the suspension was warmed to 37 °C or cooled to 4 °C. 400 µl of the suspension was added per well (0.2 mg/well) to pretreated iPSC-macrophages for each condition, and the plates were incubated for 1 hr at 37 °C or 4 °C.

Nonfluorescent and tdTomato positive *C. albicans* (clinical isolate SC5314) were grown overnight in YPD media at 30 °C with 225 rpm shaking. The cells were washed with PBS, spun at 1000 × $g$ for 5 min, and resuspended in RPMI to a density of 740,000 cells/ml. The suspensions were then warmed to 37 °C or cooled to 4 °C. 500 µl of the suspension was added per well (370,000 *C. albicans* cells) of iPSC-macrophages for each condition, and the plates were incubated for 1 hr at 37 °C or 4 °C.

After the incubation with beads, *E. coli* or *C. albicans*, the media was removed and the wells were washed with PBS. The cells were then detached with a 3 min, 37 °C incubation with trypsin, and

collected by centrifugation at 400 × $g$ for 5 min. The iPSC-macrophages incubated with beads or *E. coli* were resuspended in 200 μl of FACS buffer. The cells incubated with *C. albicans* were resuspended in PBS containing 20 μg/ml Calcofluor White (CFW) stain (Sigma), and were incubated for 15 min at RT to label unengulfed yeast. The cells were then washed with FACS buffer and pelleted at 400 G for 5 min at 4 °C, before being resuspended in 200 μl of FACS buffer.

The samples were analyzed using a FACS Fortessa SORP instrument. pHrodo Red and tdTomato fluorescence was detected through a 586/15 bandpass optical filter, on a 561 nm laser excitation. CFW was excited by a 405 nm laser, and detected through a 495LP, 525/50 bandpass optical filters. At least two replicates were done for each condition.

## Engulfment of murine apoptotic thymocytes and opsonized red blood cells

Engulfment of apoptotic cells was assayed with pHrodo-labeled mouse apoptotic thymocytes as previously described (*Miksa et al., 2009*; *Toda et al., 2012*). Briefly, thymocytes from 4- to 8-week-old C57BL/6 J mice were treated with Human leucine-zipper-tagged Fas ligand (FIZ-shFasL see below for preparation and concentration) in RPMI1640 containing 10% FBS for 2 hr at 37 °C to induce apoptosis, washed with PBS, and incubated with 0.1 μg/ml pHrodo (Invitrogen, P36600) for 30 min at RT. After the reaction was stopped with 1 ml FBS, the cells were washed with PBS containing 10% FBS and were used as prey.

Sheep red blood cells (MP Biomedicals, 0855876) were washed two times in PBS at 600 × $g$ for 5 min, at RT. The cells were resuspended in PBS, counted and further diluted to a concentration of $1 \times 10^8$ cells/ml. To opsonize the RBCs, 1/500 dilution of Rabbit Anti-Sheep RBC IgG (Cell Biolabs; 122001; CBA- 220) was added and the cells were incubated at 37 °C for 30 min. The cells were then washed twice with PBS at 600 × $g$ for 5 min, and stained in 1 ml of PBS with 10 μg/ml pHrodo under a 45-min incubation at RT. 1 ml of FBS was added to block the staining, and the cells were washed with PBS containing 10% FBS before being used in engulfment.

WT or mutant iPSCs-derived macrophages were plated at a density of 60,000 cells per well in 12-well plates and maintained in 0.75 ml of RPMI containing 10% FBS. 24 hr after plating, for select experiments (as indicted in figures) the macrophages were pretreated for 30 min with AIM100 (Tocris, 4946; 2 μM) and/or Cpd4f (Thermo Fisher Scientific, 53-100-00001; 0.5 μM), and/or R-9b (Sigma, SML2073-25MG; 4 μM) in RPMI without FBS prior to incubation with apoptotic cells or opsonized RBCs. $0.5 \times 10^6$ pHrodo-labeled apoptotic cells, or opsonized RBCs were added to 60,000 iPSCs-derived macrophages in 0.75 ml of RPMI containing 10% FBS in a 12-well plate, and incubated at 37 °C for 90 min. The cells were washed with PBS and detached with trypsin (TrypLE Express, Thermo Fisher Scientific; 12605–010). The cells were collected by centrifugation at 400 × $g$ for 5 min, suspended in 300 μl of CHES (N-cyclohexyl-2-aminoethane- sulfonic acid)–fluorescence-activated cell sorter (FACS) buffer (20 mM CHES buffer [pH 9.0] containing 150 mM NaCl and 2% FBS) and analyzed by flow cytometry with a FACS Fortessa SORP instrument. At least 5000 cells were acquired for each condition. pHrodo was excited with the yellow green laser 561 nm and detected with 610/620 nm bandpass filter.

FIZ-shFasL was produced using HEK293T as described previously (*Shiraishi et al., 2004*) and concentrated (around 50 fold) by ultrafiltration with Amicon Ultra-15 10 K column (Sigma; UFC901008). The concentration used from each batch was determined with Alexa Fluor 488 Annexin V/Dead Cell Apoptosis Kit (Invitrogen, V13241), as the concentration for which more than 80% of cells were annexin V positive and less than 10% propidium iodide positive.

## Frustrated engulfment assay and TIRF imaging

Supported lipid bilayers containing 1,2-dioleoyl-sn-glycero-3-phospho-L-serine (DOPS; Avanti Polar Lipids, 840035) were prepared as previously described (*Abeyweera et al., 2011*). Cells were incubated on bilayers (20 min, 37 °C) and fixed by adding 3% paraformaldehyde (20 min). Fixed cells were permeabilized with 0.2% Triton X-100 (15 min) and blocked in 10% goat serum/PBS (1 hr). Then cells were incubated with 0.1 U/ml Alexa Fluor 594–labeled phalloidin (Thermo Fisher Scientific; A12381; 1 hr, RT).

TIRF images of fluorescently labeled cells in contact with bilayers were collected with a 60×objective lens (1.45 NA; Olympus) using 561 nm lasers (Melles Griot) for imaging of Phalloidin. Quantification

of actin clearance ratio was performed with Matlab software (Mathworks) from the TIRF images of background-corrected mean fluorescence intensity (MFI) as previously described (*Le Floc'h et al., 2013*). Briefly, using two perpendicular linescans for each cell, the background-corrected MFI at the edges (positions F1 and F2) of the IS was compared with the background-corrected MFI of three equally spaced central positions (F3, F4, and F5) as follows: mean (F3 +F4+F5) / mean (F1 +F2). Clearance ratios derived from the two-perpendicular line- scans were averaged to yield a clearance ratio for the cell in question.

## Transcriptomic analysis by RNA-seq

70,000 iPSCs-derived macrophages were plated per well in 12-well plates in RPMI1640/GlutaMax medium supplemented with 10% FBS, and 100 ng/ml human M-CSF. WT and mutants (ACK1A156T/ K161Q and BRKG257A/G321R) iPSCs-derived macrophages were co-incubated or not with 450,000 mouse apoptotic thymocytes (as described above) in RPMI for 90 min. WT iPSC-derived macrophages were also pretreated for 30 min with AIM100 (2 μM) or cpd4f (0.5 μM) in RPMI and then incubated or not with mouse apoptotic thymocytes. Every condition was done in duplicate or triplicate. After 90 min incubation media was removed and macrophages were washed once with PBS. Then 1 ml TRIzol (Thermo Fisher Scientific; 15596018) was added per well and cells were harvested and stored at –80 °C. RNA from cells suspended in TRIzol was extracted with chloroform. Isopropanol and linear acrylamide were added, and the RNA was precipitated with 75% ethanol. Sample were resuspended in RNase-free water.

### Transcriptome sequencing

After RiboGreen quantification and quality control by Agilent Bioanalyzer, 65.8–100 ng of total RNA underwent polyA selection and TruSeq library preparation according to instructions provided by Illumina (TruSeq Stranded mRNA LT Kit; RS-122–2102), with 8 cycles of PCR. Samples were barcoded and run on a HiSeq 4000 or HiSeq 2500 in rapid mode in a 50 bp/50 bp paired end run, using the HiSeq 3000/4000 SBS Kit or HiSeq Rapid SBS Kit v2 (Illumina). An average of 50 million paired reads was generated per sample. At the most the ribosomal reads represented 7.1% of the total reads generated and the percent of mRNA bases averaged 80.4%.

### Analysis

The output data (FASTQ files, see *Table 3*) were mapped to the target genome using the rnaStar aligner (*Dobin et al., 2013*) that both maps reads to the genome and resolves reads that map across splice junctions. We used the 2 pass mapping method outlined as described (*Engström et al., 2013*), in which the reads are mapped twice. The first pass used a list of known annotated junctions from Ensemble. Novel junctions found in the first pass are then added to the list of known junctions and then a second mapping pass is done (on the second pass the RemoveNoncanoncial flag is used). After mapping we post process the output SAM files using the PICARD tools to: add read groups, AddOrReplaceReadGroups which in addition sorts the mapped reads by coordinates and coverts the file to the compressed BAM format. We then compute the expression count matrix from the mapped reads using HTSeq (www-huber.embl.de/users/anders/HTSeq) using Genecode v18 database for gene models. The raw count matrix generated by HTSeq was then processed using the R/Bioconductor package DESeq (www-huber.embl.de/users/anders/DESeq) which was used to both normalize the full dataset and analyze differential expression between sample groups.

The hypergeometric test and Gene Set Enrichment Analysis (GSEA; *Subramanian et al., 2005*) was used to identify enriched signatures using the different pathways collection in the MSigDB database (*Liberzon et al., 2011*). We used GSEA pre-ranked method from GSEA for our purpose.

## Quantitative RT-PCR

Total RNA was extracted from cells using a quick-RNA Microprep kit (Zymo research; R1050) as per manufacturer's instructions. RNA was extracted from 150,000 iPSCs-derived macrophages in 6-well plates or 37500 iPSCs-derived macrophages in 24-well plates depending on the quantity of RNA needed. Lysis buffer was added directly in the well after 1 wash with PBS, then RNA was extracted directly or cells in lysis buffer were stored at –80 °C. cDNA preparation was performed with Quantitect Reverse transcription kit (QIAGENiagen; 205313) as per manufacturer instructions. qRT-PCR

**Table 3.** RNA-Seq analysis FASTQ files.

| FASTQ File ID | Conditions |
| --- | --- |
| C12-1_IGO_08681_C13 | Ctrl |
| C12-3_IGO_08681_C15 | Ctrl |
| C12-ACs-1_IGO_08681_C_16 | Ctrl +apop cells |
| C12-ACs-2_IGO_08681_C_17 | Ctrl +apop cells |
| C12-ACs-3_IGO_08681_C_18 | Ctrl +apop cells |
| C12-AIM100-1_IGO_08681_C_19 | Ctrl +ACK1 inhib |
| C12-AIM100-2_IGO_08681_C_20 | Ctrl +ACK1 inhib |
| C12-AIM100-ACs-1_IGO_08681_C_23 | Ctrl +ACK1 inhib +apop cells |
| C12-AIM100-ACs-2_IGO_08681_C_24 | Ctrl +ACK1 inhib +apop cells |
| C12-AIM100-ACs-3_IGO_08681_C_25 | Ctrl +ACK1 inhib +apop cells |
| C12-Cpd4f-1_IGO_08681_C_21 | Ctrl +BRK inhib |
| C12-Cpd4f-2_IGO_08681_C_22 | Ctrl +BRK inhib |
| C12-Cpd4f-ACs-1_IGO_08681_C_26 | Ctrl +BRK inhib +apop cells |
| C12-Cpd41-ACs-2_IGO_08681_C_27 | Ctrl +BRK inhib +apop cells |
| F1A-1_IGO_08681_C_1 | ACK1 KO patient |
| F1A-2_IGO_08681_C_2 | ACK1 KO patient |
| F1A-3_IGO_08681_C_3 | ACK1 KO patient |
| F1A-ACs-1_IGO_08681_C_4 | ACK1 KO patient +apop cells |
| F1A-ACs-2_IGO_08681_C_5 | ACK1 KO patient +apop cells |
| F1A-ACs-3_IGO_08681_C_6 | ACK1 KO patient +apop cells |
| F9A-1_IGO_08681_C_28 | BRK KO patient |
| F9A-2_IGO_08681_C_29 | BRK KO patient |
| F9A-3_IGO_08681_C_30 | BRK KO patient |
| F9A-ACs-1_IGO_08681_C_31 | BRK KO patient +apop cells |
| F9A-ACs-2_IGO_08681_C_32 | BRK KO patient +apop cells |
| F9A-ACs-3_IGO_08681_C_33 | BRK KO patient +apop cells |
| ACs_IGO_08681_C_46 | mouse apoptotic thymocytes alone |

are done with 20 ng cDNA. qRT-PCR are performed on a Quant Studio 6 Flex using TaqMan Fast Advance Mastermix (ThermoFisher Scientific; 4444557), and TaqMan probes (ThermoFisher Scientific) for *GAPDH* (Hs02758991_m1), *TIMD4* (Hs00293316_m1), *MERTK* (Hs01031973_m1), *ITGB5* (Hs00174435_m1), *ITGB1* (Hs01127536_m1), *ITGB3* (Hs01001469_m1), *TNK2* (Hs01006880_m1), *PTK6* (Hs00966641_m1), *TNF* (Hs00174128_m1). Comparative threshold cycles (CT) was used to determine gene expression. For each sample, genes CT value were normalized with the formula $\Delta CT = CT_{gene} - CT_{GAPDH}$. For relative expression, the mean $\Delta CT$ was determined, and relative gene expression was calculated with the formula $2^{\wedge}(-\Delta CT)$.

### TNF and IL1β ELISA

hiPSC macrophages plated in 24-well tissue culture plates (density of 15,000 cells/cm$^2$) were co-cultured with or without apoptotic thymocytes (1/8 mac./apop.t. ratio) for 90 min at 37 °C in complete media (RPMI, 10% FBS, 100 ng/ml M-CSF). The wells were then washed once with PBS at 37 °C to remove the majority of remaining apoptotic thymocytes. 300 μl of complete media with or without 1 ng/ml LPS was added in each well and the cells were incubated for 18 hr at 37 °C in a $CO_2$ incubator.

The media was then collected and stored at –80 °C. ELISA was conducted according to manufacturer's protocol (Human TNF-alpha DuoSet (R&D; DY210) and IL1β DuoSet (R&D, DY201) ELISA kit).

## In vivo mouse inhibitor and pristane treatment, serum preparation, and kidney fixation

Five-week-old BALB/cByJ female mice received intra-peritoneal injection of DMSO (vehicle, 20 μl/mice), AIM100 (25 mg/kg in 20 μl), or Cpd4f (20 mg/kg in 20 μl) for 3 months. Injection were administered biweekly for the first 5 weeks, and weekly subsequently. The mice received a single intraperitoneal injection of PBS (500 μl) or pristane oil (Sigma, P9622; 500 μl) when 7 weeks old. Blood was collected by cardiac puncture. To collect serum, the blood was left to clot, undisturbed at RT for 30 min. The samples were then centrifuged at 1500 × g for 10 min at 4 °C and the supernatant/serum was collected and stored at –80 °C. The kidneys were removed, washed with PBS at 4 °C, and were fixed whole in 4% PFA overnight at 4 °C. The kidneys were then washed with PBS and place in 15% sucrose at 4 °C for 3.5 hr. The kidneys were transferred in 30% sucrose and incubated overnight at 4 °C. The kidneys were then dried with kimwipes and embedded in blocks in frozen section compound (Leica Ref:3801480). The blocks were placed in dry ice ethanol bath to freeze and were transferred to –80 °C for storage.

## Kidney histology and image analysis

Frozen kidneys were sectioned longitudinally into 12 μm sections and transferred on glass slides. The sections were left to dry for 20–30 min at RT and were transferred for storage at –20 °C. To stain, the sections were thawed at RT for 15 min in a humidified box, and rehydrated by washing in PBS at RT three times. The sections were incubated for 1 hr at RT in PBS, 5% BSA to block. The sections were then washed two times with PBS at RT and stained overnight at 4 °C with IgG AF555 (Goat Anti-Mouse; Thermo Fisher A21424) (1/500 dilution in PBS, 0.5% BSA), and Podoplanin (Syrian Hamster Anti-Mouse; Biolegend 127401; 1/100 dilution in PBS, 0.5% BSA). The sections were washed with PBS at RT three times and were incubated for 1.5 hr at RT with Anti-Syrian Hamster AF647 (Jackson, 107-605-142; 1–500 in PBS, 0.5% BSA). The sections were washed with PBS at RT three times and stain with Hoechst (Thermo Fisher 62249; 20 μM final concentration in PBS) for 10 min at RT. The slides were washed with PBS 2 times, and mounted using 170 μl of ProLong Gold antifade reagent (Invitrogen P36930) and high precision microscope cover glass (24X50 mm; 170 μM; No. 1.5 H). Entire kidneys sections were imaged on confocal microscope with ×10 objective.

To analyze IgG mean fluorescence intensity (MFI) of glomeruli, podoplanin staining was used to algorithmically generate outlines around glomeruli in full kidney section images using ImageJ. The outlines were manually verified to remove incorrectly marked glomeruli and to outline the glomeruli the algorithm missed. Over 95% of all the glomeruli in a section (about 250 per whole longitudinal kidney section) were captured and analyzed for each mice (5 mice were analyzed for each condition [30 whole kidney sections in total]). The MFI of each outlined glomeruli was extracted with ImageJ.

## Autoantigen microarray panel profiling

Autoantibody profiling was conducted by the UT Southwestern Medical Center Genomics & Microarray Core Facility (https://microarray.swmed.edu/products/product/autoantigen-microarray-service-igg-igm/). For each sample, 10 μl of serum was treated with DNAse I, diluted 1:50, and incubated with autoantigen array (Autoantigen Microarray Panel I, https://microarray.swmed.edu/products/product/autoantigen-microarray-panel-i/). The autoantibodies binding to the antigens on the array were detected with Cy3 labeled anti-IgG. The arrays were scanned with GenePix 4400 A Microarray Scanner and the images were analyzed using GenePix 7.0 software to generate GPR files. The averaged net fluorescent intensity (NFI) of each autoantigen was normalized to internal (IgG) controls. Plotted values represent Ab Scores (Log$_2$ [antigen net fluorescence intensity (NFI) x signal-to-noise ratio (SNR) +1]). Heatmaps were plotted with Morpheus (https://software.broadinstitute.org/morpheus/). Heatmap rows were sorted top to bottom starting with most significantly increased Ab Score in Cpd4f and AIM100 in comparison to DMSO treated mice. Single row p value between DMSO and Cpd4f or AIM100 mice was calculated by paired t test. Full panel p-values were calculated using a paired t test. Hierarchical clustering was done by one minus Pearson correlation with complete

linkage method. K Means clustering was done by Euclidean distance, with two clusters, with 10,000 maximum iterations.

## Quantification and statistical analysis

Data are shown as mean with individual values or as bar plots with mean values and standard deviations. Statistical significance was analyzed with GraphPad Prism unless otherwise indicated (see below). Statistical significance was determined using a Student t test, Mann-Whitney, ordinary one-way ANOVA with Tukey's multiple comparisons test, or Kruskal-Wallis multiple comparisons tests depending on the number of conditions being compared and whether the data was normally distributed as determined by a Shapiro-Wilk normality test. $p < 0.05$ values were considered statistically significant. R software was used for statistical analysis of RNA-seq data. For differential gene expression approach significance was considered for FDR (q value)$<0.05$. For gene set enrichment analysis (GSEA) approach significance was considered for FDR (q value)$<0.25$ AND p-value $<0.05$.

## Contact for reagent and resource sharing

Further information and request for reagents may be directed to and will be fulfilled by the corresponding author Frederic Geissmann: geissmaf@mskcc.org.

## Acknowledgements

We are grateful to Ingeborg Bajema and Suzanne Wilhelmus from Leiden University Medical Center and Terry Cook from Imperial College Healthcare NHS trust London for help with the pathological review of patients, and Louise Nel from Guy's and St Thomas' NHS Foundation Trust for taking care of the patients. We acknowledge the use of the MSKCC Stem Cell Research Core and MSKCC Integrated Genomics Operation Core, funded by the NCI Cancer Center Support Grant (CCSG, P30 CA08748), Cycle for Survival and the Marie-Josée and Henry R Kravis Center for Molecular Oncology. This work was supported by National Cancer Institute of the US National Institutes of Health (P30CA008748) MSKCC core grant and grants from Ludwig Institute for Cancer Research and NIH/NIAID 1R01AI130345-01 and 5R01AI124349-03 (to FG) NIH/NIAID R01-AI087644 (to MH), NIH/NCI CA58530 (to WTM), and Grants-in-Aid for Scientific Research (S) from JSPS (No. 15H05785) and Core Research for Evolutional Science and Technology from Japan Science and Technology Agency (JPMJCR14M4) (to SN). SG was supported by Fellowships from the Fondation pour la Recherche Medicale (DEA20140630127), the European Federation of Internal medicine (EFIM), the Assistance Publique-Hopitaux de Paris (Annee Recherche), and from Institut Servier. NJ was supported by Fellowships from the Arthritis Research UK Fellowship and Graham Hughes Clinical Research Fellowship.

## Additional information

### Funding

| Funder | Grant reference number | Author |
|---|---|---|
| National Cancer Institute | P30CA008748 | Frédéric Geissmann |
| National Institute of Allergy and Infectious Diseases | 1R01AI130345-01 | Frédéric Geissmann |
| National Institute of Allergy and Infectious Diseases | 5R01AI124349-03 | Frédéric Geissmann |
| National Institute of Allergy and Infectious Diseases | R01-AI087644 | Morgan Huse |
| National Cancer Institute | CA58530 | W Todd Miller |
| Grants-in-Aid for Scientific Research | No. 15H05785 | Shigekazu Nagata |

| Funder | Grant reference number | Author |
|---|---|---|
| Core Research for Evolutional Science and Technology | JPMJCR14M4 | Shigekazu Nagata |
| Fondation pour la Recherche Médicale | DEA20140630127 | Stephanie Guillet |
| European Federation of Internal Medicine | | Stephanie Guillet |
| Assistance Publique - Hôpitaux de Paris | Annee Recherche | Stephanie Guillet |
| Arthritis Research UK | Fellowship | Natasha Jordan |
| Graham Hughes Clinical Research Fellowship | | Natasha Jordan |

The funders had no role in study design, data collection and interpretation, or the decision to submit the work for publication.

## Author contributions

Stephanie Guillet, Conceptualization, Data curation, Formal analysis, Validation, Investigation, Visualization, Methodology, Writing - original draft; Tomi Lazarov, Conceptualization, Data curation, Formal analysis, Validation, Investigation, Visualization, Methodology, Writing - original draft, Project administration, Writing – review and editing; Natasha Jordan, Resources, Data curation, Investigation, Methodology; Bertrand Boisson, Data curation, Formal analysis, Supervision, Methodology, Writing – review and editing; Maria Tello, Investigation, Methodology; Barbara Craddock, Hairu Yang, Investigation; Ting Zhou, Formal analysis, Supervision, Investigation, Methodology; Chihiro Nishi, Resources, Methodology; Rohan Bareja, Nicholas D Socci, Software, Formal analysis, Methodology; Frederic Rieux-Laucat, Laurent Abel, Supervision; Rosa Irene Fregel Lorenzo, Sabrina D Dyall, Resources, Data curation, Investigation; David Isenberg, David D'Cruz, Resources, Supervision; Nico Lachmann, Supervision, Methodology; Olivier Elemento, Agnes Viale, Shigekazu Nagata, Morgan Huse, Resources, Supervision, Methodology; W Todd Miller, Resources, Formal analysis, Supervision, Visualization, Methodology; Jean-Laurent Casanova, Resources, Supervision, Methodology, Writing – review and editing; Frédéric Geissmann, Conceptualization, Supervision, Funding acquisition, Writing - original draft, Project administration, Writing – review and editing

## Author ORCIDs

Stephanie Guillet ⓘ http://orcid.org/0000-0002-9038-6765
Tomi Lazarov ⓘ https://orcid.org/0000-0002-6312-0080
Maria Tello ⓘ https://orcid.org/0000-0001-8909-6800
Shigekazu Nagata ⓘ https://orcid.org/0000-0001-9758-8426
Frédéric Geissmann ⓘ https://orcid.org/0000-0001-5029-2468

## Ethics

Human subjects: The study was approved by the Institutional Review Board of St Thomas' Hospital; Guy's hospital; the King's College London University and the Memorial Sloan Kettering Cancer Center. All subject samples were obtained after written informed consent from patients and their families according to the Helsinki convention (Ethics approval: 11/LO/1433).
Mouse studies were performed in adherence with Institutional Review Board (IACUC 15-04-006) from MSKCC.

Reviewer #1 (Public review): https://doi.org/10.7554/eLife.96085.3.sa1
Reviewer #2 (Public review): https://doi.org/10.7554/eLife.96085.3.sa2
Author response https://doi.org/10.7554/eLife.96085.3.sa3

## Additional files

### Supplementary files
• MDAR checklist

### Data availability
Genealogy trees and whole exome sequencing data of the 10 multiplex families recruited, clinical features of the 22 lupus patients recruited from 10 kindreds, and Clinical features of the patients in ACK1 and BRK families can be requested from the corresponding author. This information is not included in the manuscript due to patient confidentiality considerations. Raw data files for the RNA sequencing analysis have been deposited in the NCBI Gene Expression Omnibus under accession number GSE118730. Further information and request for reagents may be directed to and will be fulfilled by the corresponding author Frederic Geissmann (geissmaf@mskcc.org).

The following dataset was generated:

| Author(s) | Year | Dataset title | Dataset URL | Database and Identifier |
|---|---|---|---|---|
| Guillet S, Geissmann F | 2021 | Response of human iPSCs-derived macrophages to apoptotic cells, role of ACK1 and BRK tyrosine kinases | https://www.ncbi.nlm.nih.gov/geo/query/acc.cgi?acc=GSE118730 | NCBI Gene Expression Omnibus, GSE118730 |

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
