## [Editor Report · eLife assessment]

In this **important** study, the authors found, with the use of statistical methods, that compound heterozygous rare deletion variants affecting the kinase-domain of non-receptor tyrosine kinase TNK2/ACK1 and PTK6/BRK are associated with human systemic lupus erythematosus (SLE). The authors use a **convincing** mouse experimental model and human-induced pluripotent stem cell (hiPSC)-derived macrophages to clarify cause-effect relationships and the cellular basis of nephritis. With the identification of new SLE-related genes, this manuscript improves our understanding of human SLE pathogenesis.

---

## [Referee Report · Reviewer #1 (Public review)]

The authors report compound heterozygous deleterious variants in the kinase domains of the non-receptor tyrosine kinases (NRTK) TNK2/ACK1 in familial SLE. They suggest that ACK1 and BRK deficiencies are associated with human SLE and impair efferocytosis.

The experiments in this revision showing that a weekly injection of ACK1 or BRK inhibitors induced various kinds of lupus-related autoantibodies in BALB/c supported the pivotal role of ACK1/BRK in systemic autoimmunity, although treated mice failed to demonstrate the full picture of lupus.

---

## [Referee Report · Reviewer #2 (Public review)]

In this manuscript, the authors revealed that genetic deficiencies of ACK1 and BRK are associated with human SLE. First, the authors found that compound heterozygous deleterious variants in the kinase domains of the non-receptor tyrosine kinases (NRTK) TNK2/ACK1 in one multiplex family and PTK6/BRK in another family. Then, by an experimental blockade of ACK1 or BRK in a mouse SLE model, they found an increase in glomerular IgG deposits and circulating autoantibodies. Furthermore, they reported that ACK and BRK variants from the SLE patients impaired the MERTK-mediated anti-inflammatory response to apoptotic cells in human induced pluripotent stem cells (hiPSC)-derived macrophages. This work identified new SLE-associated ACK and BRK variants and a role for the NRTK TNK2/ACK1 and PTK6/BRK in efferocytosis, providing a new molecular and cellular mechanism of SLE pathogenesis.

---

## [Author Response]

The following is the authors’ response to the original reviews.

**Public Reviews:**

**Reviewer #1 (Public Review):**
Summary:The authors report compound heterozygous deleterious variants in the kinase domains of the non-receptor tyrosine kinases (NRTK) TNK2/ACK1 in familial SLE. They suggest that ACK1 and BRK deficiencies are associated with human SLE and impair efferocytosis.Strengths:The identification of similar mutations in non-receptor tyrosine kinases (NRTKs) in two different families with familial SLE is a significant finding in human disease. Furthermore, the paper provides a detailed analysis of the molecular mechanisms behind the impairment of efferocytosis caused by mutations in ACK1 and BRK.Weaknesses:A critical point in this paper is whether the loss of function of ACK1 or BRK contributes to the onset of familial SLE. The authors emphasize that inhibitors of ACK1/BRK worsened IgG deposition in the kidneys in a pristane-induced SLE model, which contributes not to the onset but to the exacerbation of SLE, thus only partially supporting their claim.

The evidence supporting that the loss of function of ACK1 or BRK contributes to the onset of SLE in the patients from the 2 families mostly relies on the genetic analysis. As the reviewer states, the observation that inhibitors of ACK1/BRK worsened IgG deposition in the kidneys in a pristane-induced SLE model supports the genetic evidence.

To further address the possible role of ACK1 or BRK variants in the onset of autoimmunity in vivo*,* we treated wild-type (WT) BALB/cByJ female mice with inhibitors in the absence of pristane.

The results indicated that mice that had received a weekly injection of ACK1 or BRK inhibitors developed a large array of serum anti-nuclear IgG antibodies, including but not limited to autoantibodies associated with SLE such as anti-histones, anti-chromatin, anti U1-snRNP, anti-SSA, and anti-Ku in comparison to the control group inhibitor treated mice (Revised Fig 3A). However, they did not develop glomerular deposit of IgG after 12 weeks of treatment, in contrast to mice that have received Pristane (Revised Fig. 3B,C, Figure 3-figure supplement 1).

These additional data suggests that inhibition of ACK1 and BRK stimulates the production of serum autoantibodies, which strengthen the claim that ACK1 and BRK kinase deficiency contribute to autoimmunity in BALB/cByJ.

**Reviewer #2 (Public Review):**
Summary:In this manuscript, the authors revealed that genetic deficiencies of ACK1 and BRK are associated with human SLE. First, the authors found that compound heterozygous deleterious variants in the kinase domains of the non-receptor tyrosine kinases (NRTK) TNK2/ACK1 in one multiplex family and PTK6/BRK in another family. Then, by an experimental blockade of ACK1 or BRK in a mouse SLE model, they found an increase in glomerular IgG deposits and circulating autoantibodies. Furthermore, they reported that ACK and BRK variants from the SLE patients impaired the MERTK-mediated anti-inflammatory response to apoptotic cells in human induced pluripotent stem cells (hiPSC)-derived macrophages. This work identified new SLE-associated ACK and BRK variants and a role for the NRTK TNK2/ACK1 and PTK6/BRK in efferocytosis, providing a new molecular and cellular mechanism of SLE pathogenesis.Strengths:This work identified new SLE-associated ACK and BRK variants and a role for the NRTK TNK2/ACK1 and PTK6/BRK in efferocytosis, providing a new molecular and cellular mechanism of SLE pathogenesis.Weaknesses:Although the manuscript is well-organized and clearly stated, there are some points below that should be considered:In this study, the authors used forward genetic analyses to identify novel gene mutations that may cause SLE, combined with GWAS studies of SLE. To further explore the importance of these variants, haplotype analysis of two candidate genes could be performed, to observe the evolution and selection relationship of candidate genes in the population (UK 1000 biobank, for example).

To investigate whether ACK1/TNK2 or BRK/PTK6 were subject to selection, we gathered data using different metrics quantifying negative selection in the human genome. We collected the f parameter from SnIPRE1, lofTool2, and evoTol3, as well as intraspecies metrics from RVIS4, LOEUF5, and pLI6 (including pRec). We also used our in-house CoNeS metric7. None of these indicators suggest that the genes are under strong negative selection (Revised Figure 2-figure supplement 2). This is consistent with the deficiency being recessive. We also tested the variants with a MAF greater than 0.005. We found them to be neutral. We therefore did not test whether they were associated with any phenotype in the UK Biobank.

Although the authors focused on SLE and macrophage efferocytosis in their studies, direct evidence of how macrophage efferocytosis significantly affects SLE is lacking. This point should at least be explicitly introduced and discussed by citing appropriate literature.

We provide a more detailed description of the role of macrophage efferocytosis in autoimmunity and SLE in the revised manuscript. Specifically, we state (in the results section, paragraph: ACK1 and BRK kinase domain variants may lose the ability to link MERTK to RAC1, AKT and STAT3 activation for efferocytosis): “NRTKs such as ACK1 8 and PTK2/FAK 9 are also downstream targets of the TAM family receptor MERTK which is expressed on macrophages and controls the anti-inflammatory engulfment of apoptotic cells, a process known as efferocytosis 10-12. Efferocytosis allows for the clearance of apoptotic cells before they undergo necrosis and release intracellular inflammatory molecules, and simultaneously leads to increased production of anti-inflammatory molecules (TGFb, IL-10, and PGE2) and a decreased secretion of proinflammatory cytokines (TNF-alpha, IL-1b, IL-6) 10-14. In line with these findings, mice deficient in molecular components used by macrophages to efficiently perform efferocytosis, such as MFG-E8, MERTK, TIM4, and C1q, develop phenotypes associated with autoimmunity10,11,14-27. Furthermore, defects in efferocytosis are also observed in patients with SLE and glomerulonephritis14,28-31.“

It is still not clear how the target molecules identified in this paper may influence macrophage efferocytosis. More direct evidence should be established.

Our studies show that wt -but not variants- of ACK1 and BRK are activated by MERTK, a key receptor that mediates the recognition of apoptotic cells. Our studies also show that wt -but not variants- activate RAC1 which is necessary for engulfment and phosphorylate AKT and STAT3 which are involved in the anti-inflammatory response to PtdSer recognition.

The TAM family receptor MERTK mediates recognition of PtdSer on apoptotic cells via GAS6 and Protein S 10,15,32 leading to their engulfment, which involves activation of RAC1 for actin reorganization and the formation of a phagocytic cup 9,33. Using IP kinase assays we show that MERTK and GAS6 can activate the kinase activity of wild-type ACK1 8 or BRK but not of the patient’s ACK1 or BRK variant alleles (Figure 4D). To further support the role of ACK1 and BRK downstream from PtdSer recognition and uptake of apoptotic cells, we show that reference ACK1 and BRK alleles, in contrast to the patient variant alleles, can activate RAC1 to generate RAC-GTP which is necessary for engulfment 9,33 (Figure 4C).

PtdSer recognition also typically stimulates an anti-inflammatory process mediated in part via AKT 34 and STAT3 and their target genes such as SOCS3 35-41 and results in the inhibition of LPS-mediated production of inflammatory mediators such as TNF and IL-1b, and the production of cytokines such as IL-10, TGFb 11,25-27,42. Consistent with this literature and the findings of the paper, we show that reference ACK1 and BRK, unlike the patient’s variant alleles, can phosphorylate AKT and STAT3 (Figure 4A, B). The role of ACK1 and BRK in these signaling pathways is further supported by our transcriptomics data comparing the response of controls, patients, and inhibitor-treated iPSC-derived macrophages to apoptotic thymocytes by RNA-seq. Specifically, we show Transcriptional repressors including the AKT targets ATF3, TGIF1, NFIL3, and KLF4, the STAT3 targets SOCS3 and DUSP5, as well as CEBPD and the inhibitor of E-BOX DNA Binding ID3 were among the top-ten genes which expression is induced by apoptotic cells in WT macrophages (Figure 4F), but this regulation was lost in mutant and inhibitor-treated macrophages (Figure 4F).

For some transcriptional repressors mentioned in their studies, the authors should check whether there is clear experimental evidence. If not, it is recommended to supplement the experimental verifications for clarity.

Transcriptional repressors including the AKT targets ATF3, TGIF1, NFIL3, and KLF4, the STAT3 targets SOCS3 and DUSP5, as well as CEBPD and the inhibitor of E-BOX DNA Binding ID3 were among the top-ten genes which expression is induced by apoptotic cells in WT macrophages (Figure 4F), but this regulation was lost in mutant and inhibitor-treated macrophages (Figure 4F).

In the manuscript we cited published evidence, to the best of our knowledge, for the role of these genes in the regulation of inflammatory responses. Specifically we state: “ATF3, TGIF1, NFIL3, and KLF4 are involved in the negative regulation of inflammation in macrophages 35-38, SOCS3 is an inhibitor of the macrophage inflammatory response and DUSP5 is a negative regulator of ERK activation 39,40,43. These data suggest that the kinase domain of ACK1 and BRK contribute to the macrophage anti-inflammatory gene expression program driven by apoptotic cells.”

In Figures 4C and 4D, it is seen that the usage of inhibitors causes cytoskeletal changes, however this reviewer would not have expected such large change. Did the authors check whether the cells die after heavy treatment by the inhibitors?

We carefully examine the viability of Isogenic WT, BRK and ACK1 mutant macrophages (left panel) and of WT macrophages treated with ACK1 or BRK inhibitors and we did not observed changes in viability (Figure 4-figure supplement 2).

**Recommendations for the authors:**

**Reviewer #1 (Recommendations For The Authors):**
A crucial step in the development of SLE is the production of autoantibodies. It is shown in Figure 2F that inhibitors of ACK1/BRK enhanced the production of autoantibodies against histones and SSA in a pristane-induced SLE model, which is a significant result that could support the authors' claim. Strangely, this autoantigen panel does not include double-stranded DNA, RNP, or Sm, which should be presented regarding antibody production.

We thank the reviewer for this comment. In the revised manuscript (Revised Figure 3 – Supplement 1) we added the remainder of the autoantibody panel, which includes double-stranded DNA, RNP, and Sm autoantibody levels. We also added the results for serum IgG autoantibody levels in BALB/cByJ mice treated for three months with DMSO, ACK1, or BRK inhibitors but did not receive a pristane injection (Revised Figure 3A). This data shows that mice which received ACK1 or BRK inhibitors had increased serum IgG autoantibodies in comparison to DMSO treated controls.

Additionally, if there is information that inhibitors of ACK1/BRK promote the differentiation of follicular helper T cells, memory B cells, and plasma cells in a pristane-induced SLE model, it could be considered indirect evidence supporting the authors' claims.

These are not available at present to the best of our knowledge.

**Reviewer #2 (Recommendations For The Authors):**
Minor points:* In the literature, unpaired t-tests and ordinary one-way ANOVA (Tukey's multiple comparisons test) were used for statistical analysis, which requires data to be normally distributed. This part of the proposal is reflected in the text, and the non-conforming results need to be statistically analyzed using the non-parametric test of graphpad prism.

We would like to thank the reviewer for pointing out this oversight. In the revised manuscript, for all applicable datasets, we tested whether the data was normally distributed using a Shapiro-Wilk normality test. For datasets that were normally distributed statistical significance was determined by a Student t test or ordinary one-way ANOVA with Tukey’s multiple comparisons test depending on the number of conditions being compared and the experimental setup. In contrast, for datasets that were not normally distributed statistical significance was determined using a Mann-Whitney, Kruskal-Wallis multiple comparisons tests, or Wilcoxon matched-pairs signed rank test depending on the experimental setup. P values below 0.05 were considered significant for all statistical tests.

The authors used different methods to represent the level of significant difference. Therefore, it is suggested that the significance level should be expressed by letters.

As suggested by the reviewer, in the revised manuscript we have designated the significance level throughout all figures using letters (p, or q values).

For RNA-seq, more information should be provided in the paper. For example, the correlation between sample biological replicates, the total number of differentially expressed genes, and randomly selected genes for qRT-PCR results verification.

We would like to thank the reviewer for pointing out this oversight. In the revised manuscript we provided more information regarding the RNA-seq dataset, including a Principal Component Analysis (PCA) showing correlation between sample replicates (Revised Figure 4-figure supplement 1A), as well as a table indicating the number of upregulated and downregulated genes between relevant datasets (Revised Figure 4-figure supplement 1B).

The results of the RNA-seq analysis indicated that ACK1 and BRK contribute to the macrophage anti-inflammatory gene expression program driven by apoptotic cells. MERTK-dependent anti-inflammatory program elicited by apoptotic cells on macrophages is best evidenced by the reduction of LPS-mediated production of inflammatory mediators such as TNF or IL1b 25-27,34,44. Therefore, to validate the RNA-seq results in a functional manner we tested the decrease of LPS-induced production of TNF and IL1b by apoptotic cells in isogenic WT, ACK1 deficient, and BRK deficient macrophages. Consistent with the RNA-seq data, the functional assays indicated that ACK1 and BRK kinase activities are required for the decrease of TNF and IL1b production induced by LPS in response to apoptotic cells (Revised Figure 4H,I).

The raw data files for the RNA-seq analysis have been deposited in the NCBI Gene Expression Omnibus under accession number GEO: GSE118730.

The authors did not have the formats for some of the citations correct. This should be fixed.

References were reformatted.

(1) Eilertson, K. E., Booth, J. G. & Bustamante, C. D. SnIPRE: selection inference using a Poisson random effects model. *PLoS Comput Biol* 8, e1002806 (2012). https://doi.org/10.1371/journal.pcbi.1002806

(2) Fadista, J., Oskolkov, N., Hansson, O. & Groop, L. LoFtool: a gene intolerance score based on loss-of-function variants in 60 706 individuals. *Bioinformatics* 33, 471-474 (2017). https://doi.org/10.1093/bioinformatics/btv602

(3) Rackham, O. J., Shihab, H. A., Johnson, M. R. & Petretto, E. EvoTol: a protein-sequence based evolutionary intolerance framework for disease-gene prioritization. *Nucleic Acids Res* 43, e33 (2015). https://doi.org/10.1093/nar/gku1322

(4) Petrovski, S., Wang, Q., Heinzen, E. L., Allen, A. S. & Goldstein, D. B. Genic intolerance to functional variation and the interpretation of personal genomes. *PLoS Genet* 9, e1003709 (2013). https://doi.org/10.1371/journal.pgen.1003709

(5) Karczewski, K. J. *et al.* The mutational constraint spectrum quantified from variation in 141,456 humans. *Nature* 581, 434-443 (2020). https://doi.org/10.1038/s41586-020-2308-7

(6) Lek, M. *et al.* Analysis of protein-coding genetic variation in 60,706 humans. *Nature* 536, 285-291 (2016). https://doi.org/10.1038/nature19057

(7) Rapaport, F. *et al.* Negative selection on human genes underlying inborn errors depends on disease outcome and both the mode and mechanism of inheritance. *Proc Natl Acad Sci U S A* 118 (2021). https://doi.org/10.1073/pnas.2001248118

(8) Mahajan, N. P., Whang, Y. E., Mohler, J. L. & Earp, H. S. Activated tyrosine kinase Ack1 promotes prostate tumorigenesis: role of Ack1 in polyubiquitination of tumor suppressor Wwox. *Cancer Res* 65, 10514-10523 (2005). https://doi.org/10.1158/0008-5472.CAN-05-1127

(9) Wu, Y., Singh, S., Georgescu, M. M. & Birge, R. B. A role for Mer tyrosine kinase in alphavbeta5 integrin-mediated phagocytosis of apoptotic cells. *J Cell Sci* 118, 539-553 (2005). https://doi.org/10.1242/jcs.01632

(10) Scott, R. S. *et al.* Phagocytosis and clearance of apoptotic cells is mediated by MER. *Nature* 411, 207-211 (2001). https://doi.org/10.1038/35075603

(11) Henson, P. M. & Bratton, D. L. Antiinflammatory effects of apoptotic cells. *J Clin Invest* 123, 2773-2774 (2013). https://doi.org/10.1172/JCI69344

(12) Henson, P. M. Cell Removal: Efferocytosis. *Annu Rev Cell Dev Biol* 33, 127-144 (2017). https://doi.org/10.1146/annurev-cellbio-111315-125315

(13) deCathelineau, A. M. & Henson, P. M. The final step in programmed cell death: phagocytes carry apoptotic cells to the grave. *Essays Biochem* 39, 105-117 (2003). https://doi.org/10.1042/bse0390105

(14) Nagata, S. Apoptosis and Clearance of Apoptotic Cells. *Annu Rev Immunol* 36, 489-517 (2018). https://doi.org/10.1146/annurev-immunol-042617-053010

(15) Cohen, P. L. *et al.* Delayed apoptotic cell clearance and lupus-like autoimmunity in mice lacking the c-mer membrane tyrosine kinase. *J Exp Med* 196, 135-140 (2002). https://doi.org/10.1084/jem.20012094

(16) Hanayama, R. *et al.* Autoimmune disease and impaired uptake of apoptotic cells in MFG-E8-deficient mice. *Science* 304, 1147-1150 (2004). https://doi.org/10.1126/science.1094359

(17) Miyanishi, M., Segawa, K. & Nagata, S. Synergistic effect of Tim4 and MFG-E8 null mutations on the development of autoimmunity. *Int Immunol* 24, 551-559 (2012). https://doi.org/10.1093/intimm/dxs064

(18) Colonna, L., Parry, G. C., Panicker, S. & Elkon, K. B. Uncoupling complement C1s activation from C1q binding in apoptotic cell phagocytosis and immunosuppressive capacity. *Clin Immunol* 163, 84-90 (2016). https://doi.org/10.1016/j.clim.2015.12.017

(19) Nagata, S., Hanayama, R. & Kawane, K. Autoimmunity and the clearance of dead cells. *Cell* 140, 619-630 (2010). https://doi.org/10.1016/j.cell.2010.02.014

(20) Kimani, S. G. *et al.* Contribution of Defective PS Recognition and Efferocytosis to Chronic Inflammation and Autoimmunity. *Front Immunol* 5, 566 (2014). https://doi.org/10.3389/fimmu.2014.00566

(21) Hanayama, R., Tanaka, M., Miwa, K., Shinohara, A., Iwamatsu, A. & Nagata, S. Identification of a factor that links apoptotic cells to phagocytes. *Nature* 417, 182-187 (2002). https://doi.org/10.1038/417182a

(22) Kawano, M. & Nagata, S. Lupus-like autoimmune disease caused by a lack of Xkr8, a caspase-dependent phospholipid scramblase. *Proc Natl Acad Sci U S A* 115, 2132-2137 (2018). https://doi.org/10.1073/pnas.1720732115

(23) Watanabe-Fukunaga, R., Brannan, C. I., Copeland, N. G., Jenkins, N. A. & Nagata, S. Lymphoproliferation disorder in mice explained by defects in Fas antigen that mediates apoptosis. *Nature* 356, 314-317 (1992). https://doi.org/10.1038/356314a0

(24) Singer, G. G., Carrera, A. C., Marshak-Rothstein, A., Martinez, C. & Abbas, A. K. Apoptosis, Fas and systemic autoimmunity: the MRL-lpr/lpr model. *Current opinion in immunology* 6, 913-920 (1994).

(25) Cvetanovic, M. & Ucker, D. S. Innate immune discrimination of apoptotic cells: repression of proinflammatory macrophage transcription is coupled directly to specific recognition. *J Immunol* 172, 880-889 (2004). https://doi.org/10.4049/jimmunol.172.2.880

(26) Fadok, V. A., Bratton, D. L., Konowal, A., Freed, P. W., Westcott, J. Y. & Henson, P. M. Macrophages that have ingested apoptotic cells in vitro inhibit proinflammatory cytokine production through autocrine/paracrine mechanisms involving TGF-beta, PGE2, and PAF. *J Clin Invest* 101, 890-898 (1998). https://doi.org/10.1172/JCI1112

(27) Voll, R. E., Herrmann, M., Roth, E. A., Stach, C., Kalden, J. R. & Girkontaite, I. Immunosuppressive effects of apoptotic cells. *Nature* 390, 350-351 (1997). https://doi.org/10.1038/37022

(28) Herrmann, M., Voll, R. E., Zoller, O. M., Hagenhofer, M., Ponner, B. B. & Kalden, J. R. Impaired phagocytosis of apoptotic cell material by monocyte-derived macrophages from patients with systemic lupus erythematosus. *Arthritis Rheum* 41, 1241-1250 (1998). https://doi.org/10.1002/1529-0131(199807)41:7<1241::AID-ART15>3.0.CO;2-H

(29) Baumann, I. *et al.* Impaired uptake of apoptotic cells into tingible body macrophages in germinal centers of patients with systemic lupus erythematosus. *Arthritis Rheum* 46, 191-201 (2002). [https://doi.org/10.1002/1529-0131(200201)46:1](https://doi.org/10.1002/1529-0131(200201)46:1<191::AID-ART10027>3.0.CO;2-K)

(30) Schrijvers, D. M., De Meyer, G. R. Y., Kockx, M. M., Herman, A. G. & Martinet, W. Phagocytosis of apoptotic cells by macrophages is impaired in atherosclerosis. *Arterioscl Throm Vas* 25, 1256-1261 (2005). https://doi.org/10.1161/01.ATV.0000166517.18801.a7

(31) Morioka, S., Maueroder, C. & Ravichandran, K. S. Living on the Edge: Efferocytosis at the Interface of Homeostasis and Pathology. *Immunity* 50, 1149-1162 (2019). https://doi.org/10.1016/j.immuni.2019.04.018

(32) Seitz, H. M., Camenisch, T. D., Lemke, G., Earp, H. S. & Matsushima, G. K. Macrophages and dendritic cells use different Axl/Mertk/Tyro3 receptors in clearance of apoptotic cells. *J Immunol* 178, 5635-5642 (2007). https://doi.org/10.4049/jimmunol.178.9.5635

(33) Mao, Y. & Finnemann, S. C. Regulation of phagocytosis by Rho GTPases. *Small GTPases* 6, 89-99 (2015). https://doi.org/10.4161/21541248.2014.989785

(34) Sen, P. *et al.* Apoptotic cells induce Mer tyrosine kinase-dependent blockade of NF-kappaB activation in dendritic cells. *Blood* 109, 653-660 (2007). https://doi.org/10.1182/blood-2006-04-017368

(35) Vergadi, E., Ieronymaki, E., Lyroni, K., Vaporidi, K. & Tsatsanis, C. Akt Signaling Pathway in Macrophage Activation and M1/M2 Polarization. *J Immunol* 198, 1006-1014 (2017). https://doi.org/10.4049/jimmunol.1601515

(36) Byles, V. *et al.* The TSC-mTOR pathway regulates macrophage polarization. *Nat Commun* 4, 2834 (2013). https://doi.org/10.1038/ncomms3834

(37) Liao, X. *et al.* Kruppel-like factor 4 regulates macrophage polarization. *J Clin Invest* 121, 2736-2749 (2011). https://doi.org/10.1172/JCI45444

(38) Roberts, A. W., Lee, B. L., Deguine, J., John, S., Shlomchik, M. J. & Barton, G. M. Tissue-Resident Macrophages Are Locally Programmed for Silent Clearance of Apoptotic Cells. *Immunity* 47, 913-927 e916 (2017). https://doi.org/10.1016/j.immuni.2017.10.006

(39) Matsukawa, A. *et al.* Stat3 in resident macrophages as a repressor protein of inflammatory response. *J Immunol* 175, 3354-3359 (2005).

(40) Sica, A. & Mantovani, A. Macrophage plasticity and polarization: in vivo veritas. *J Clin Invest* 122, 787-795 (2012). https://doi.org/10.1172/JCI59643

(41) Yi, Z., Li, L., Matsushima, G. K., Earp, H. S., Wang, B. & Tisch, R. A novel role for c-Src and STAT3 in apoptotic cell-mediated MerTK-dependent immunoregulation of dendritic cells. *Blood* 114, 3191-3198 (2009). https://doi.org/10.1182/blood-2009-03-207522

(42) Rothlin, C. V., Carrera-Silva, E. A., Bosurgi, L. & Ghosh, S. TAM receptor signaling in immune homeostasis. *Annu Rev Immunol* 33, 355-391 (2015). https://doi.org/10.1146/annurev-immunol-032414-112103

(43) Seo, H. *et al.* Dual-specificity phosphatase 5 acts as an anti-inflammatory regulator by inhibiting the ERK and NF-kappaB signaling pathways. *Sci Rep* 7, 17348 (2017). https://doi.org/10.1038/s41598-017-17591-9

(44) Camenisch, T. D., Koller, B. H., Earp, H. S. & Matsushima, G. K. A novel receptor tyrosine kinase, Mer, inhibits TNF-alpha production and lipopolysaccharide-induced endotoxic shock. *J Immunol* 162, 3498-3503 (1999).